# Mesoscale phase separation of chromatin in the nucleus

**Gaurav Bajpai[†]\*, Daria Amiad Pavlov[‡], Dana Lorber[‡], Talila Volk, Samuel Safran\***

Weizmann Institute of Science, Rehovot, Israel

**Abstract** Intact-organism imaging of *Drosophila* larvae reveals and quantifies chromatin-aqueous phase separation. The chromatin can be organized near the lamina layer of the nuclear envelope, conventionally fill the nucleus, be organized centrally, or as a wetting droplet. These transitions are controlled by changes in nuclear volume and the interaction of chromatin with the lamina (part of the nuclear envelope) at the nuclear periphery. Using a simple polymeric model that includes the key features of chromatin self-attraction and its binding to the lamina, we demonstrate theoretically that it is the competition of these two effects that determines the mode of chromatin distribution. The qualitative trends as well as the composition profiles obtained in our simulations compare well with the observed intact-organism imaging and quantification. Since the simulations contain only a small number of physical variables we can identify the generic mechanisms underlying the changes in the observed phase separations.

**\*For correspondence:**
gaurav.bajpai@weizmann.ac.il
(GB);
sam.safran@weizmann.ac.il (SS)

**Present address:** [†]Department
of Chemical and Biological
Physics, Weizmann Institute of
Science, Rehovot, Israel;
[‡]Department of Molecular
Genetics, Weizmann Institute of
Science, Rehovot, Israel

**Competing interests:** The
authors declare that no
competing interests exist.

**Reviewing editor:** Karsten
Kruse, University of Geneva,
Switzerland

## Introduction

Chromatin is a complex, linear macromolecule comprising DNA and histone proteins which in eukaryotic cells, is localized in the nucleus where it is solubilized in water, salts, and other small molecules (*Cooper and Hausman, 2000*; *Phillips et al., 2012*). In many studies, chromatin organization in interphase is homogeneous on the nuclear scale. In this 'conventional' picture, the chromatin and the aqueous solvent uniformly fill the nucleus as a single phase (*Rosa and Shaw, 2013*). Even the conventional picture accounts for phase separation similar to that of soluble AB block copolymers (*Rubinstein and Colby, 2003*), with regions of transcriptionally active euchromatin (A block) separated from regions of relatively inactive heterochromatin (B block); however, both are assumed to be homogeneously solubilized in the aqueous phase (*Erdel and Rippe, 2018*; *Narlikar, 2020*; *Strom et al., 2017*; *Larson et al., 2017*). For example, Hi-C experiments reveal such AB chromatin compartmentalization (*Lieberman-Aiden et al., 2009*) but do not provide information about their location within the nucleus.

In most previous measurements, nuclear-scale phase separation of the chromatin and the aqueous phase has not been considered. However, recent super-resolution microscopy reveals a 'marshland' of chromatin and aqueous phase, with a non-uniform distribution of chromatin at the submicron level (*Cremer et al., 2015*). Another study observed a larger scale phase separation of chromatin and the aqueous phase in early development, with the chromatin localized to the nuclear periphery (*Popken et al., 2014*). Both these observations did not distinguish the A and B blocks (eu and hetero chromatin) and instead identified large regions of DNA-rich chromatin separated from DNA-poor regions, presumably of the aqueous phase. More recently, we have presented (see *Appendix 1—figure 1*) intact-organism imaging of *Drosophila* larvae nuclei where the chromatin was labeled by H2B-RFP, and the nuclear envelope was labeled by Nesprin/Klar-GFP. These experiments reveal and quantify chromatin-aqueous phase separation and its control by changes in the nuclear volume and the interaction of chromatin with the lamina (part of the nuclear envelope) at the nuclear periphery (*Amiad-Pavlov et al., 2020*). Here, we demonstrate theoretically that it is the competition of these two effects, together with the self-attraction of the chromatin to itself that

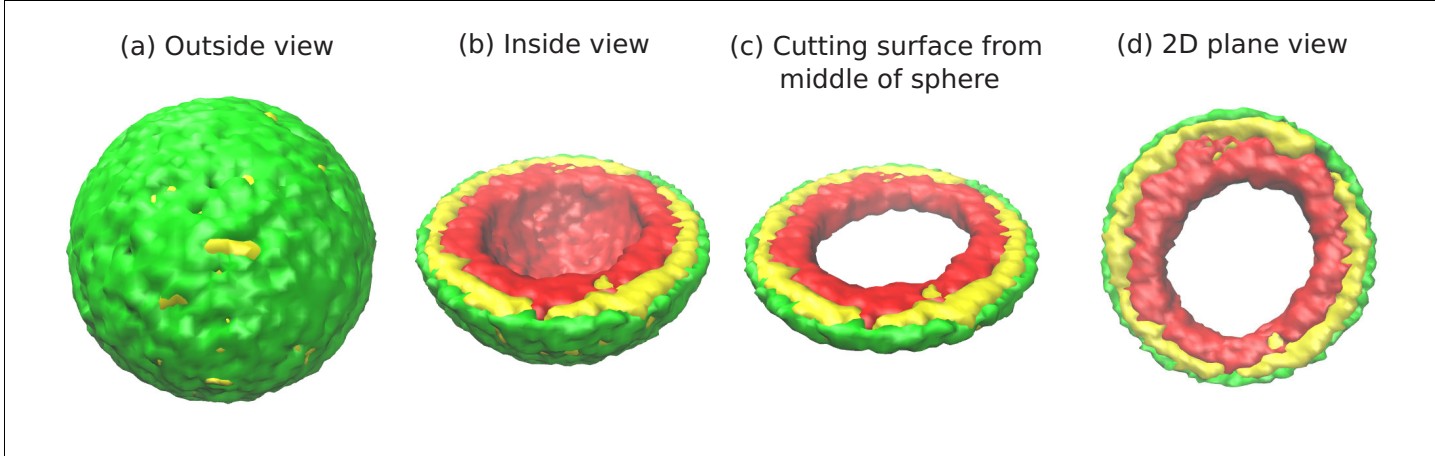

**Figure 1.** Snapshots of the simulated system from different views. This representation is generated by drawing the surfaces around the beads that represent the lamin and chromatin. The same set of pictures shown here are depicted in a different graphical representation, in terms of the beads themselves, in *Figure 1—figure supplement 1*. (a) *Outside view:* Our model, spherical nucleus is enclosed by lamin (NL) beads (green). Within the sphere, the chromatin chain of N = 37,333 beads contains two types of beads: LAD (yellow) and non-LAD (red). (b) *Inside view:* The sphere is cut at the equatorial plane to reveal one hemisphere, so that chromatin (red and yellow) is visible. The central region of the nucleus is devoid of chromatin for the particular conditions ($\phi = 0.3, \psi = 1, \epsilon = 1$) of this simulation. (c) *Cutting a slice near the equatorial plane:* A slice is cut from the hemisphere, resulting in a 3D surface which has width of $(1/10)$ of the sphere diameter. (d) *2D planar view:* We show the equatorial (xy) plane view of the 3D slice and observe that the central regions contains no chromatin.

The online version of this article includes the following figure supplement(s) for figure 1:

**Figure supplement 1.** Snapshots of the simulated system visualized in terms of beads.

determine whether the chromatin is conventionally or peripherally distributed within the nucleus; we also show that more complex organizational modes are also possible. The model we present below that focuses on the concentration profile of chromatin (*whether A or B*) within the nucleus, is appropriate when the chromatin self-attraction relative to the chromatin-aqueous phase interaction, as well as the chromatin-lamina interaction are larger than the difference between the AA and BB interactions. The qualitative trends as well as the compositional profiles obtained in our simulations compare well with the observed, intact-organism imaging and quantification. Since the simulations contain only a small number of physical variables, this allows us to identify the generic mechanisms underlying the changes in the phase separation that are observed.

The physical insight obtained here focuses on the competition of two primary interactions of chromatin within the nucleus: (i) chromatin-lamina, which is attractive and tends to organize the chromatin peripherally (ii) chromatin-chromatin, that if attractive, tends to condense the chromatin and separate it from the aqueous phase. The nuclear envelope contains lamina which is a dense, fibrillar network of proteins that provide anchoring points for various proteins that bind to chromatin on one end and the lamina on the other (*Ulianov et al., 2019*; *Kind et al., 2013*; *Yáñez-Cuna and van Steensel, 2017*; *Meister et al., 2010*). These binding proteins associate with lamins that, in general, comprise two A-type (lamin A and lamin C) and two B-type (lamin B1 and lamin B2) proteins (*Moir et al., 2000*). However, *Drosophila* has only one A-type lamin (lamin C) and one B-type lamin (lamin Dm0) gene (*Riemer et al., 1995*; *Schulze et al., 2009*). A-type and B-type lamins are found at the nuclear periphery while A-type lamins are also found in the nuclear interior (*Briand and Collas, 2020*; *Naetar et al., 2017*). Those chromatin domains, which bind to the lamins are called lamina associated domains (LADs) which are similarly divided in two groups: A-LAD and B-LAD corresponding to interactions with A-type lamin and B-type lamin, respectively (*van Steensel and Belmont, 2017*; *Briand and Collas, 2020*). The anchor protein lamin B receptor (LBR) binds lamin B and acts as a tether between chromatin and the nuclear lamina (*Solovei et al., 2013*). Experiments performed in the absence of LBR observed a loss of peripheral LADs and an inverted architecture with LADs localized within the nuclear interior (*Solovei et al., 2013*; *Briand and Collas, 2020*). The role of lamin Dm0 in *Drosophila* is the same as LBR in mammalian cells (*Wagner et al., 2004*; *Ulianov et al., 2019*).

Previous models of chromatin-lamina interactions resolved euchromatin and heterochromatin domains in the context of conventional organization of chromatin (*Chiang et al., 2019*; *Sati et al., 2020*; *Falk et al., 2019*) where phase separation with the aqueous phase is not considered. Another model of chromatin and lamina interactions, treats chromatin as a self-avoiding polymer in good solvent and focuses on the role of lamina in the formation of chromosome territories (*Maji et al., 2020*). However, if chromatin were a self-avoiding polymer (with no self-attraction), its radius of gyration in aqueous solution would typically be larger than the diameter of the nucleus. Thus, under confinement, chromatin would fill the entire volume of nucleus, which is not consistent with the observation (*Amiad-Pavlov et al., 2020*) of peripheral or central chromatin organization. In contrast, the study presented here illustrates in an intuitive manner, how chromatin-lamina interactions control peripheral and other organizational modes that show chromatin-aqueous phase separation. The experimental observations (see *Appendix 1—figure 1*) of peripheral organization demonstrate the condensation of chromatin in the outer part of the nucleus, indicating the presence of chromatin self-attraction (*Amiad-Pavlov et al., 2020*). This motivates our model which focuses on attractive interactions within the chromatin so that it acts in the nucleus as a polymer in a relatively bad solvent. For large enough nuclear volumes and strong enough chromatin-lamina attractions, instead of filling the nucleus, the chromatin is condensed in only part of the volume near the nuclear periphery.

The model we study is generic in nature, short-ranged and does not depend on the detailed molecular origin of the chromatin self-attraction. This is an appropriate approach for understanding and predicting the nuclear-scale concentration profile of the chromatin and aqueous regions in vivo, where the identity and function of all the molecular actors are so far unknown. Nevertheless, there is ample reason to believe that such attractions are present and important in determining chromatin organization. For example, the positively charged histone tails attract negative DNA linkers and can thus promote condensation of chromatin (*Gibson et al., 2019*; *Bajpai and Padinhateeri, 2020*; *Hancock, 2007*). Protein condensates that interact with chromatin can also result in self-attraction. A prominent example is the phase separation of HP1 in the nucleus, which binds to the heterochromatin domains of chromatin (*Strom et al., 2017*; *Larson et al., 2017*). If one protein can bridge two or more chromatin regions that are close in physical space, but possibly far along the stretched chromatin chain, this can also result in an effective chromatin-chromatin attraction. Other nuclear proteins that interact with chromatin can play a similar role. For example, nucleoplasmic lamin A can act to 'crosslink' two chromatin regions in vivo (*Bronshtein et al., 2016*; *Bronshtein et al., 2015*; *Stephens et al., 2017*). Other studies have indeed considered chromatin to act as a polymer gel cross-linked by non-histone protein complexes (*Biggs et al., 2019*; *Irianto et al., 2013*).

In this paper, we systematically examine how the interplay between chromatin-lamina interactions, intra-chromatin interactions, and hydration affect the chromatin organization. The biological observations of different organizational modes are simulated and explained by the polymeric properties of chromatin and the competition of its bulk interactions with the surface interactions with the lamina. Different organizational modes can affect gene expression since different amounts and sections of the chromosome are exposed to the aqueous phase and the proteins and enzymes that it contains. This may be why changes in nuclear volume have been demonstrated to result in changes in gene expression (*Jain et al., 2013*; *Thomas et al., 2002*). Our simulation results also provide suggestions to experimental groups on how to vary the minimal number of parameters that will experimentally result in different modes of chromatin organization in vivo.

## Model

As explained above, we are interested in the *generic* physical effects that determine the chromatin concentration profile at the nuclear scale. However, to predict this from computer simulations of chromatin as a self-attracting, confined polymer that is also attracted to the lamina at the nuclear periphery, we must treat a specific polymeric model. We therefore fixed the polymer monomer size, molecular weight, and persistence length, and instead varied the physical properties of the nucleus, including its volume and chromatin-lamina interactions. This was motivated by the intact-organism experiments (*Amiad-Pavlov et al., 2020*). Comparing the live to fixed cells showed a reduction in the nuclear volume by a factor of about three and a transition from peripheral (live) to conventional (fixed) chromatin organization. Lamin C overexpression resulted in a transition from peripheral (WT) to central (and wetting droplet) organization in the mutant (see *Appendix 1—figure 1*). In our

simulations, we also varied the magnitude of the chromatin self-attraction to demonstrate the difference in organization of chromatin in a good solvent vs. a poor one.

## Chromatin chain

For the understanding of the generic properties of interest here, we treated a single chromosome. Aqueous phase separation in case of multiple chromosomes can be mapped from our results by using the appropriate chromatin volume fraction. We were motivated by the X chromosome of *Drosophila* since its distribution of LAD domains is known. We therefore modeled chromatin as a polymer using a bead-spring model (*Naumova et al., 2013*) with $N = 37,333$ beads. Each bead represents three nucleosomes (around 600 bp of DNA) whose diameter is denoted by $\sigma = 10$ nm. Thus, our model chromosome contains about 22.4Mbp, comparable to the X chromosome. The bead-spring interactions ($E_s^{chrom}$) and persistence length (bending energy $E_b^{chrom}$) are determined in the standard manner as detailed in Materials and methods, where it is shown that the appropriate persistence length is two beads in our representation.

In addition to the spring-bead model that accounts for the connectivity of nearest neighbor beads along the polymer chain, we include short-range interactions (both attractive and repulsive) between *any* two beads that are close enough in 3D space; they do not necessarily have to be 'close' along the chain. These, non-bonding interactions between any two beads were taken using a Lennard-Jones (LJ) potential,

$$E_{LJ} = \begin{cases} 4\epsilon \sum_{\substack{i,j \\ i<j}} \left[ \left( \frac{\sigma}{r_{ij}} \right)^{12} - \left( \frac{\sigma}{r_{ij}} \right)^6 \right] & \text{when } r_{ij} < r_c \\ 0 & \text{when } r_{ij} \geq r_c \end{cases} \tag{1}$$

where $r_c$ refers to a cutoff distance beyond which LJ interaction is set to zero. Here $r_{ij}$ is the distance in 3D space between $i^{th}$ and $j^{th}$ beads and $\epsilon$ is strength of potential. If there are no attractive forces between the beads (but only steric, hard-core repulsions), we truncate the LJ potential at the distance at which the repulsive force goes to zero which gives $r_c = 2^{1/6}\sigma$. For the case of attractive forces, of particular interest in our work, the cutoff distance is taken as $r_c = 2.5\sigma$. For $\epsilon = 1k_BT$, the chromosome is a self-avoiding chain when $r_c = 2^{1/6}\sigma$ and a self-attractive one when $r_c = 2.5\sigma$ (see Appendix 2 and *Appendix 2—figure 1* for scaling exponent results).

## Nuclear confinement and volume

We model the nucleus to which the chromatin is localized, as spherical shell of confinement radius $R_c$ and define a confinement potential $E_{LJ}^{wall}$ that accounts for the hard-core repulsion between the chromatin beads and the spherical wall. Each bead of the chromatin chain interacts with its nearest point on the wall through the potential $E_{LJ}$ with LJ strength $1k_BT$ and the nuclear-chromatin cut off distance $r_n = 2^{1/6}\sigma$. This is the distance at which the repulsive force between the chromatin beads and the nuclear shell falls to zero. Motivated by the experiments where the nuclear volumes for live nuclei are significantly larger than those of fixed nuclei, we allow for variations of the confinement volume (with constant chromatin volume given by fixing $N$ and the bead size). We define the parameter $\phi$, as the global volume fraction of chromatin within the nucleus, with $(1 - \phi)$, the volume fraction of the aqueous phase:

$$\phi = \frac{\text{Volume of chromatin chain}}{\text{Volume of confinement}} = \frac{N \times \frac{4}{3}\pi(\sigma/2)^3}{\frac{4}{3}\pi R_c^3} \tag{2}$$

We varied $R_c$ from 210 nm to 360 nm and hence $\phi$ from 0.1 to 0.5, where small values of $\phi$ model the case of chromatin in hydrated nuclei (aqueous solution) and large value of $\phi$ is for chromatin in dehydrated nuclei (*Li et al., 2017*; *Maeshima et al., 2020*; *Pawley, 2006*).

## Chromatin LAD and non-LAD domains

To account for the biophysics of the bonding of the LAD domains of chromatin to the lamina, our chain comprises two types of beads in which only the LAD beads can be bonded to the lamina. The remaining non-LAD beads do not form such bonds. In order to simulate a specific system, we

modeled the 22.4 Mbp regions of chromosome X (ChrX) of *Drosophila* where the distribution of the LAD regions is available online (*Ho et al., 2014*). That data indicates that 48% of the sequences are LAD with an average domain size of 90 kbp. We analyzed the data which shows an alternating pattern of LAD and non-LAD along the chromosome length (see *Appendix 3—figure 1*) and used these patterns in our simulations. To demonstrate that our coarse-grain model does not depend on the size of the chromosome and the locations of the LADs along the chromosome, we also used a Monte-Carlo method to randomly distribute the LAD regions along the chain (see details of method and results in Appendix 4, *Appendix 4—figures 1*, and *Appendix 4—figures 2*).

## Chromatin-chromatin interactions

At the microscopic level, there are many possible interactions (e.g. electrostatic attraction between DNA and histone-tails, phase separation of the DNA binding non-histone protein HP1, and DNA crosslinking by nucleoplasmic lamin-A) which can cause chromatin self-attraction and lead to its separation from the aqueous phase. Since there may be additional bridging proteins and because even the ones we listed are not known in detail, we use a generic model for short-range chromatin-chromatin attractions. As outlined above, we model chromatin self-attraction by an attractive LJ potential (with cutoff $r_c = 2.5\sigma$ as explained above). In the section below that considers variations of the interaction strength $\epsilon$, we first consider the case where all beads (both LAD and non-LAD) interact with the same LJ potential. In addition, in the Appendix 7, we also consider the case in which interactions between two LAD beads are more strongly attractive compared with the interactions of two non-LAD beads or a LAD non-LAD pair. The reason for considering this more specific case, is to present an analogy for the micro phase separation of hetero and eu chromatin.

## Chromatin-lamina interactions

To model the lamina that interacts with the LAD domains of the chromatin, we introduce additional beads which are localized to the confinement surface. For simplicity, we modeled the lamin beads which represent the thin, laminar shell, as static. The very short-range LAD-lamina bonding is mediated by specific proteins which we account for below. However, for the LAD regions of the chromatin to kinetically find the lamina after the initial conditions of our simulations, we include an additional, somewhat longer range attraction of the lamin and LAD beads, modeled by a relatively weak, LJ potential with $\epsilon_{lm} = 1k_BT$ with cutoff of $r_c = 2.5\sigma$ (as explained above), where σ is the minimum hard-core distance between LAD and lamina beads.

The interactions of the non-LAD beads with the lamina are purely repulsive (due to their excluded volume) and to model that we use a LJ interaction with $\epsilon_{ln} = 1k_BT$ and cutoff distance $r_c = 2^{1/6}\sigma$, at

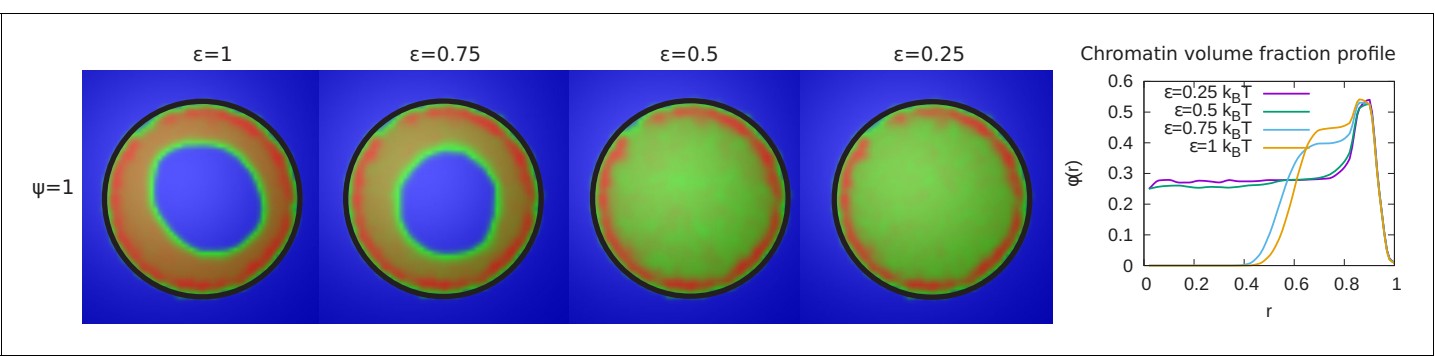

**Figure 2.** Variation of the chromatin concentration profile as a function of the intra-chromatin attractive interactions. *Left panel:* Chromatin concentrations are shown for different intra-chromatin attraction strengths ($\epsilon$) with a volume fraction of chromatin $\phi = 0.3$ and maximal LAD-lamina interactions ($\psi = 1$). For smaller values of the attractions, the chromatin no longer shows peripheral localization, and fills the entire nucleus. This demonstrates the role of the chromatin self-attractions in stabilizing peripheral chromatin organization with a relatively high local volume fraction (red) of chromatin compared to the case of relatively small self-attractions where the chromatin fills the entire nucleus, with a smaller local volume fraction (green). *Right panel:* The local volume fraction profiles of chromatin for different $\epsilon$ plotted as a function of the radial distance $r$ where $r = 0$ is the nuclear center and $r = 1$ is the position of the nuclear envelope. These local volume fraction profiles are obtained by averaging over the azimuthal and polar angles in thin spherical shells at distances from the nuclear center to the nuclear periphery.

which point the repulsive force falls to zero. The attractive LAD-lamin attraction results in localization of the LAD near periphery compared with the non-LAD. The total, non-bonding (physical) interaction energy between lamin and chromatin ($E_{LJ}^{\mathrm{lamina}}$) is calculated from the sum of these individual LAD and non-LAD LJ interactions with lamin.

However, the most important feature of the chromatin-lamina interaction is the bonding of the LAD and the lamina via specific proteins. This depends on the availability of these proteins and the nature of their binding to the lamina. In the experiments (*Amiad-Pavlov et al., 2020*) that induced lamin C overexpression, there were significant changes in the chromatin organization with much less bonding of the chromatin to the lamina. Lamin overexpression has been suggested to repress the bonding activity of LBR proteins that bind the LAD domains to lamin B (*Buxboim et al., 2017*; *Cho et al., 2017*). These qualitative changes in chromatin organization as a function of the fraction of bonds that can be formed, cannot be studied using the LJ interaction which results in polymer adsorption to the lamina and not bonding, which can be restricted due to a finite number of binding sites or available binding proteins. We therefore allow for the possibility that not all LAD domains can bind to the lamina by introducing the parameter $0 \leq \psi \leq 1$ which represents the fraction of the LAD domains that can bind to the lamina (for example, due to a reduction in the number of binding proteins relative to the number of LAD).

When $\psi = 1$ all LAD can bond to the lamin beads while $\psi = 0$ represents the situation in which no LAD-lamina bonds are possible (e.g. due to a lack of bonding proteins or bonding sites within the lamina). We allow for bond formation when a LAD bead is within a distance of $r = r_b = 2.5\sigma$ from a lamin bead where the LJ interaction is attractive. A bond is then formed with an energy $E_s^{lamina} = -k_{bond}(r_b - \sigma)^2 + k_{bond}(r - \sigma)^2$, where we take $k_{bond} = 10 k_B T/\sigma^2$. Thus, when the LAD-lamin distance, $r$ is equal to $r_b$, the bond energy is zero and when $r = \sigma$, the optimal bond energy, is attained. When the distance is either larger or smaller than σ, the energy due to the stretching of the spring is positive; at distances smaller than σ, the LJ repulsion dominates due to the excluded volume. In our model, a single LAD bead can form a maximum of one bond with a given lamin bead. The difference between the LJ interactions of the lamin and LAD and the bonding energy is seen when one considers the detachment of the lamin and LAD. For the LJ interaction, there are no energy barriers and the probabilities of all the distances between the LAD/lamin are just related to the Boltzmann factor (*Blake and De Coninck, 2011*) of the LJ potential. For bonding of the LAD and lamin (e.g. through the LBR protein, not explicitly considered in our model), a more general approach is used that does not require a description of all the intermediate configurations between bonding and non-bonding. LAD-lamin bonds are formed when the distance $r$ between a LAD bead and the lamin obeys $r < r_b$ where we take $r_b = 2.5\sigma$. Once the bond is formed, the LAD-lamin interaction is given by a spring with an equilibrium spring length of σ and spring constant of $10 k_B T/\sigma^2$. The bonds can break only if $r$ fluctuates to values greater than $r_b$ and the breaking occurs with a probability $P_{break} = e^{-U_{bond}/k_B T}$ where for a situation of detailed balance, $U_{bond}$ is the bond formation energy which we choose to be $10 k_B T$. (In Appendix 5, we describe the effects of variations of the cutoff distance for bond breaking.)

Adding up all these contributions yields the total potential energy of chromatin in our model nucleus:

$$E_{tot} = E_s^{chrom} + E_b^{chrom} + E_{LJ}^{chrom} + E_{LJ}^{wall} + E_{LJ}^{lamina} + E_s^{lamina} \tag{3}$$

The total potential energy depends on the positions of the beads that determine their mutual interactions and those of the LAD beads with the lamina; the force on a given bead arises from the gradient of this potential energy.

## Results

We now summarize the results of the Brownian dynamics simulations and discuss how conventional, peripheral and central organization of chromatin can be achieved by changing hydration (chromatin volume fraction $\phi$), chromatin-lamina interactions (fraction of bonded LAD $\psi$), and intra-chromatin interactions (attraction strength $\epsilon$). The simulations below demonstrate and quantify how the competition of chromatin self-attraction, hydration, and LAD-lamina attraction compete to determine the qualitatively different chromatin concentration profiles shown below. This is summarized at the end

of this section in a state diagram showing the transitions between the various nuclear-scale chromatin organizational modes. The comparison with the experimental trends is addressed in the Discussion section.

## Peripheral organization of chromatin

Before presenting the transitions in chromatin organization as a function of the parameters discussed above, we first discuss one important example of peripheral organization in pictorial detail. Motivated by the experiments, we tune the simulation parameters $\phi, \psi$, and $\epsilon$ to the range where the equilibrium organization of the chromatin is peripheral, for example: $\phi = 0.3, \psi = 1, \epsilon = 1$. In *Figure 1*, we present snapshots of simulations which demonstrate that for these particular values of $(\phi, \psi, \epsilon)$ the chromatin is localized near the nuclear periphery. We next systematically vary these important physical parameters and show how the simulation results change to reveal different organizational modes of the chromatin.

## Variation of intra-chromatin attractive interactions

In our simulations, 48% of the chromatin are LAD domains (corresponding to $\psi = 1$) which remain strongly bound to the lamina. For a chromatin volume fraction $\phi = 0.3$ and maximal chromatin-lamina attraction (i.e. all LAD domains can bond to the laminar, $\psi = 1$), we varied the value of the chromatin self-attraction, $\epsilon$, which is the same for all pairs of beads (LAD pairs, non-LAD pairs, and LAD/non-LAD pairs). The LAD-lamin interaction strength and the chromatin volume fraction were held fixed. In the left panel of *Figure 2*, we depict the chromatin concentrations in the equatorial, xy-plane calculated by taking the average of many frames (snapshots), while the system remained in equilibrium. The chromatin concentrations are shown with different colors where blue is chromatin-free and thus represents the aqueous phase, green represents low chromatin concentrations and red, high concentrations. When the intra-chromatin attractive interaction strength is relatively large, $\epsilon = 1$, the chromatin organization is peripheral. When the self-attraction is sufficiently small ($\sim \epsilon < 1/2$), the chromatin fills the entire volume, showing conventional organization. In the right hand panel of *Figure 2*, we plot the local volume fraction of chromatin $\phi(r)$, within a spherical shell as a function of the normalized radial distance $r$, where $r = 0$ represents the nuclear center and $r = 1$ represents the position of the nuclear envelope. We show the results for different values of the chromatin self-attraction. Here, $\phi(r)$ is the local volume fraction of chromatin; the average of $\phi(r)$ over the spherical volume is the global volume fraction $\phi$. In the plot, we observe that for higher intra-chromatin attraction ($\epsilon \geq 0.75$), the local volume fraction shows a peak near the nuclear periphery ($r = 1$) and decreases to zero near the center, consistent with peripheral organization. For smaller intra-chromatin attraction ($\epsilon \leq 0.5$), the local volume fraction has a peak near the nuclear periphery ($r = 1$) but never decreases to zero since, in conventional organization, the chromatin fills the entire volume of the nucleus. This demonstrates the role of the chromatin self-attractions in stabilizing peripheral chromatin organization with a relatively high local volume fraction (red) of chromatin compared to the case of relatively small self-attractions where the chromatin fills the entire nucleus, with a smaller local volume fraction (green).

## Variation of the fraction of the LAD domains bound to the lamina

In our simulations, we considered the case in which not all LAD domains (in our simulations, 48% of the chromatin) can bind to the lamin beads. This was suggested by the experiments in systems where lamin C was overexpressed and may be due to a reduction in the number of lamin-binding proteins or the obscuring or burial of lamin-binding sites in the overexpressed situation (*Buxboim et al., 2017*). The parameter ψ represents the fraction of LAD beads that can potentially bind to the lamin. We now present the results of simulations that vary the value of ψ for fixed values of $\phi = 0.3$ and $\epsilon = 1$. The left hand panel of *Figure 3*, shows that a decrease in the value of ψ results in a transition of the chromatin organization from peripheral to central. While in the equatorial plane, this is indeed mostly central organization, the 3D structure is more complex with a 'droplet' of chromatin which 'wets' and thus contacts the bottom of the hemisphere due to the LAD-lamin attractions. This 'wetting droplet' configuration is discussed in more detail and shown in Appendix 5 and *Appendix 5—figure 1*, where its kinetic features are also explored. A similar progression is observed for a small chromatin volume fraction, $\phi = 0.1$, and shown in *Figure 3—figure supplement*

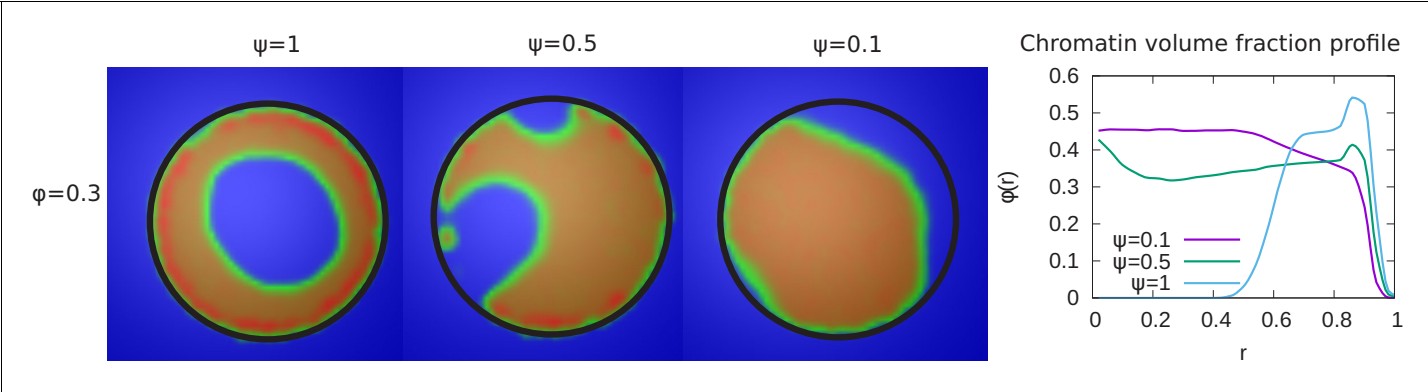

**Figure 3.** Variation of the chromatin concentration profile as a function of the fraction of LAD domains bound to the lamina. *Left panel:* Variation in the fraction $\psi$ of LAD beads that can bond to the lamin, associated domains relative to its maximal value of $\psi = 1$, where 48% of the chromatin consists of LAD domains that can bind to the lamin. The values of the chromatin volume fraction $\phi = 0.3$ and chromatin self-interaction strength, $\epsilon = 1$ are fixed. Those LAD domains (a fraction, $1 - \psi$) not bound to the lamin are not necessarily found near the periphery of the nucleus and are mixed with the non-LAD chromatin. For small values of $\psi$, the chromatin is no longer peripherally localized but fills the nucleus more uniformly (but see the discussion of the 'wetting droplet' in Appendix 5 and *Appendix 5—figure 1*), since there are relatively few LAD-lamin bonds to localize the chromatin at the nuclear periphery. *Right panel:* The local volume fraction profiles of chromatin as a function of the radial distance $r$, where $r = 0$ is the nuclear center and $r = 1$ is the location of the nuclear envelope.

The online version of this article includes the following figure supplement(s) for figure 3:

**Figure supplement 1.** Transition from peripheral to central chromatin localization.

*1*. In the right hand panel of *Figure 3*, the local chromatin volume fraction shows a peak near the nuclar periphery ($r = 1$) for $\psi = 1$ while for $\psi = 0.1$ the peak of the local volume fraction is shifted toward the center ($r = 0$).

## Effect of hydration (variation of chromatin volume fraction) of the nucleus

In the experiments, the volume of live *Drosophila* larva nuclei had an average value of $1183 um^3$, while fixed nuclei had an average volume of $381 um^3$, with corresponding changes in the chromatin organization from peripheral to conventional. This motivated our simulation study of the effect of changes in the relative fractions of the chromatin on chromatin organization, by changing the parameter $\phi$. For relatively high chromatin volume fractions ($\phi = 0.5$), chromatin fills the entire volume of nucleus and shows conventional organization; there is no phase separation of chromatin and aqueous phase due to the overabundance of chromatin in the system (left panel of *Figure 4*). As we decrease the

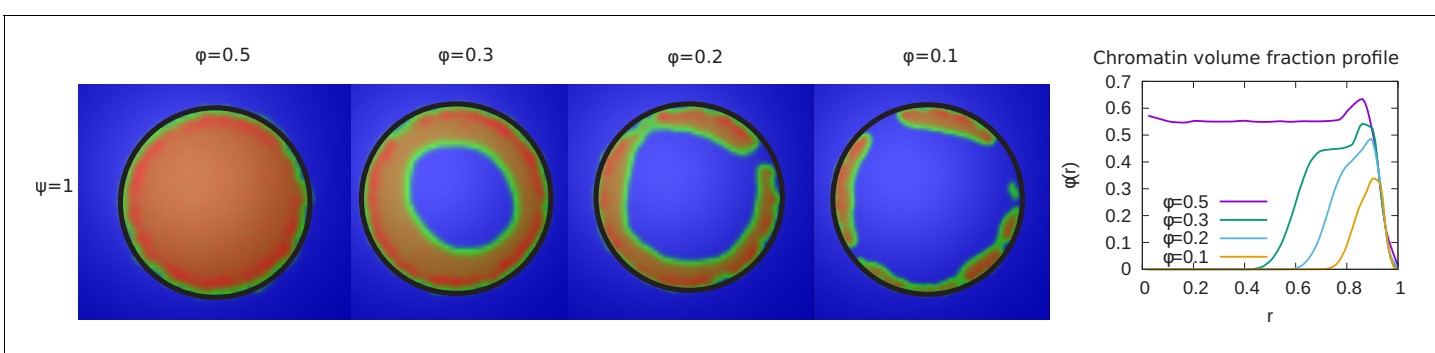

**Figure 4.** Variation of the chromatin concentration profile as a function of the chromatin volume fraction. *Left panel:* Chromatin concentrations are shown for different volume fractions of chromatin, $\phi$. *Right panel:* The local volume fraction profiles show peripheral organization for $\phi = 0.3$ and $\phi = 0.1$ but not for $\phi = 0.5$. This demonstrates the transition from peripheral to conventional chromatin organization as the nucleus is dehydrated and $\phi$ is increased.

volume fraction of chromatin, the organization shows a transition from conventional to one that is more peripheral. In this case, there is sufficient aqueous phase so that phase separation is possible.

### Transitions from peripheral to uniform to central localization of chromatin

In the results shown above, we see how chromatin organization changes when we change $\phi, \psi$ and $\epsilon$. Here, we summarize the transitions of chromatin organization from peripheral to central or from peripheral to uniform via a state diagram. We did many simulations (total 84 simulations) for different values of the parameters $(\phi, \psi, \epsilon)$ and calculated local volume fraction profiles for these ranges. From the plots of local volume fraction, we classified the chromatin organization as peripheral, central (which can include the wetting droplet configurations discussed in Appendix 5 and *Appendix 5—figure 1*) or conventional. In *Figure 5*, we present the state diagram showing the transitions in chromatin organization by variations of the chromatin volume fraction, fraction of LAD that can bond to the lamin, and the intra-chromatin attraction strength $(\phi, \psi, \epsilon)$. These results show how this minimal set of coarse-grained, simulation parameters $(\phi, \psi, \epsilon)$ already determines a rich variety of chromatin organization in the nucleus.

## Discussion

We have shown here that the observed transitions in nuclear-scale chromatin organization in eukaryotic cells can be understood from the physics of a simple polymer model. The comparison of the theoretical and experimental trends indicates that these transitions are due to competition of polymer entropy, self-attraction of the chromatin and the chromatin-lamina attraction. By varying the values of the three generic parameters $(\phi, \epsilon, \psi)$ that respectively control these effects, our simulations demonstrated the transitions between peripheral, central and conventional organization of chromatin as seen in the experiments.

For example, peripheral organization of chromatin was observed in our simulations when: (i) The radius of confinement within the nucleus is greater than radius of gyration of chromatin ($R_c > R_g$ where

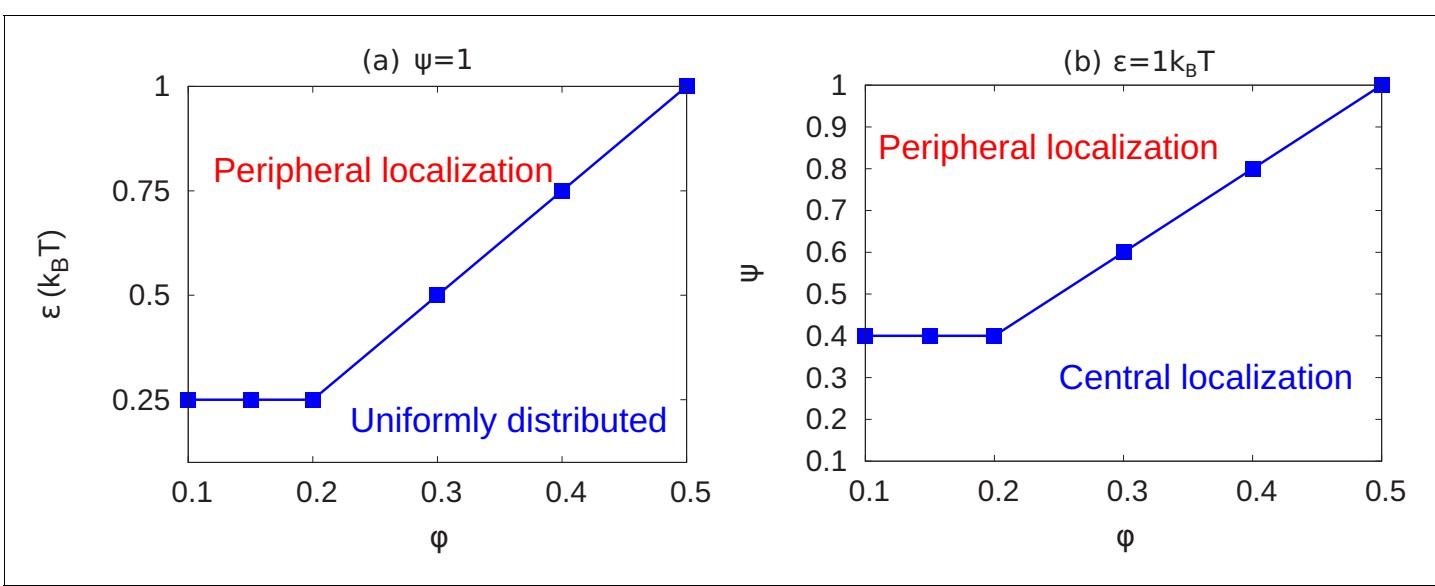

**Figure 5.** State diagrams showing the transition from peripheral to uniform to central localization of chromatin. (a) For a fixed value of $\psi = 1$ (maximally bonded LAD), we calculated the local volume fraction obtained from simulations with different pairs (24 pairs) of the chromatin volume fraction and self-attraction $(\phi, \epsilon)$ where $0.1 \leq \phi \leq 0.5$ and $0.25 \leq \epsilon \leq 1$. For each pair of $(\phi, \epsilon)$, we used the plots of the local volume fraction to determine the chromatin organization mode. In the graph, the blue line shows the transition between conventional (uniformly distributed) chromatin organization, and peripheral organization. (b) For a fixed value of the chromatin self-attraction $\epsilon = 1 k_B T$, we calculated the local volume fraction obtained from simulations with different pairs (60 pairs) of the chromatin volume fraction and fraction of LAD that can bond to the lamina $(\phi, \psi)$ where $0.1 \leq \phi \leq 0.5$ and $0.1 \leq \psi \leq 1$. The blue line shows the transition from central (and wetting drop) to peripheral chromatin organization. In figures (a) and (b), the bars are the simulation results while the line is a guide to the eye.

$R_g$ is for the self-attractive case), which is relevant for relatively small volume fractions of chromatin ($\phi = 0.3$). (ii) Chromatin must be self-attractive for its localization on the nuclear periphery, which occurs for significant chromatin self-attraction ($\epsilon = 1$);. (iii) Chromatin-lamina interactions are strong, as in the simulations for the case where all (or most) of the LADs can bond to the lamina ($\psi = 1$) (see *Figure 1*).

Starting with these parameter values that resulted in peripheral organization, we then varied $\phi, \psi, \epsilon$ to determine from our simulations, the transitions from peripheral to central to conventional, by varying one parameter at a time. We found transitions from peripheral to conventional when the self-attraction and volume fractions were decreased: $0.25 \leq \epsilon \leq 1$ and $0.1 \leq \phi \leq 0.5$ (see *Figure 2* and *Figure 4*). We also found transitions from peripheral to central by reducing the chromatin-lamina interactions ($0 \leq \psi \leq 1$) (see *Figure 3*). We plotted the state diagrams for these transitions in *Figure 5*. These qualitatively track the observed experimental trends, for example, peripheral to conventional chromatin organization when the cell is dehydrated so that the chromatin volume fraction is increased. In Appendix 1 and Appendix 5, we discuss in more detail the comparison of the simulations and the experimental images that are published elsewhere (*Amiad-Pavlov et al., 2020*). In Appendix 6 and *Appendix 6—figure 1*, we calculated contact map from our simulation results (whose snapshots are shown in *Appendix 1—figure 1a*) and compared the contact maps of peripheral, conventional and central organization. The results show that chromatin in peripheral organization is condensed in a manner similar to conventional organization but is more condensed compared with central (*Appendix 6—figure 1a* and (b)) organization. We have also shown that lamins prevent chromosomes from mixing and help chromosomes remain in well-defined regions (analogous to territories) (see *Appendix 6—figure 1c*). We have also shown that like conventional, peripheral chromatin is also organized in a fractal manner (see *Appendix 6—figure 1d*).

An interesting experimental feature observed in the peripheral organization of chromatin, is that euchromatin (active chromatin, typically associated with non-LAD) and heterochromatin (inactive chromatin, typically associated with LAD) regions do not separate in the radial directions, but are separated in the perpendicular (angular) direction (see *Figure 6c*). Our simulations show that this can be the case even if the self-attraction of LAD and non-LAD are the same; this corresponds to the

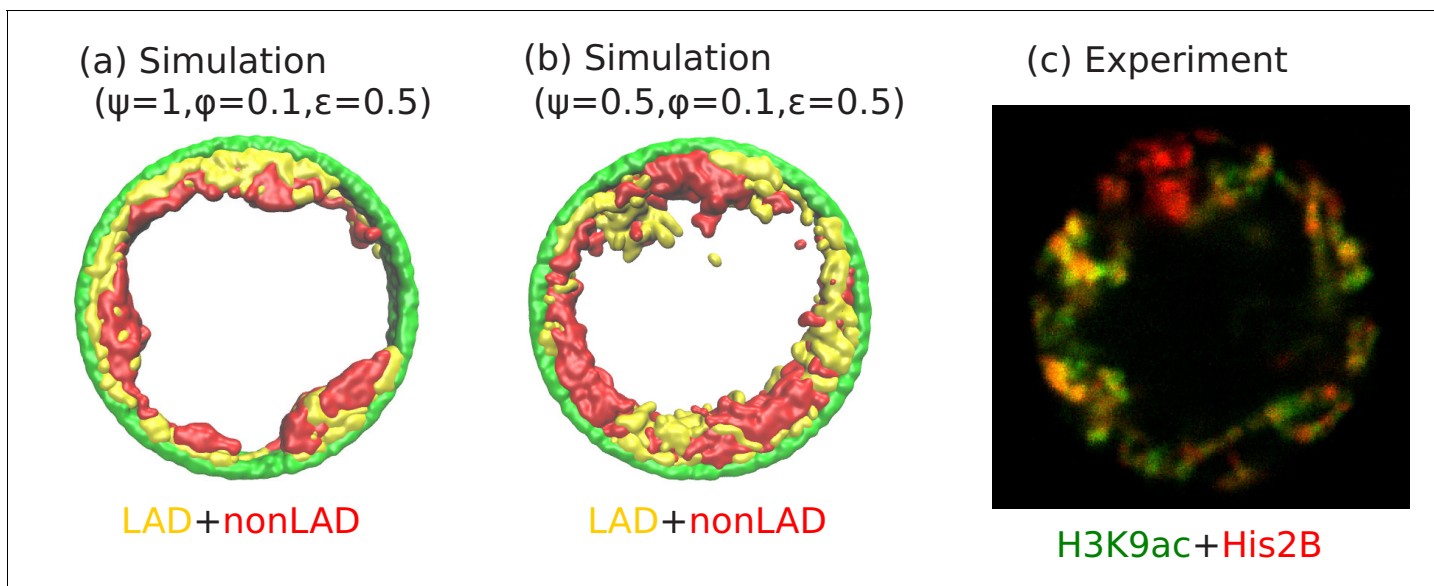

**Figure 6.** Angular separation of LAD and non-LAD chromatin. (**a**) Simulation snapshot for parameter values $\psi = 1$, $\phi = 0.1$, and $\epsilon = 0.5$, shows peripheral organization with LAD near lamina and non-LADs separated from LADs in the radial direction. (**b**) Simulation snapshot for parameter values $\psi = 0.5$, $\phi = 0.1$, and $\epsilon = 0.5$, shows peripheral organization with alternating LAD and non-LAD regions in the angular direction at the nuclear periphery. (**c**) Experimentally labeled H3K9ac (euchromatin/most of non-LAD/green) and His2B (chromatin/LAD and non-LAD/red) in muscle nuclei of intact *Drosophila* larvae, shows heterochromatin (associated with LAD) by dark red color and euchromatin (associated with non-LAD) by merging the red and green colors in peripheral organization. Both the experiments and simulations show an angular distribution of LAD and non-LAD as opposed to a radial distribution.

limit of negligibly small differences in heterochromatin/euchromatin interactions relative to their interactions with the aqueous phase and with the lamina. In this case, the angular separation originates in the stronger binding of the LAD domains to the lamina. We observed the angular separation of LAD and non-LAD in our simulations, when we allowed both the LAD and the non-LAD to have LJ attractions to the lamina (arising from physical interactions such as van der Waals) but allow only the LAD to additionally forms biochemical bonds with the lamina (Biophysically, this corresponds to bonds formed via proteins such as BAF). The angular alternation of LAD and non-LAD on the nuclear periphery occurs when only a fraction of the LAD can bind to the lamina (e.g., due to a limited number of binding proteins or sites) corresponding to $\psi = 0.5$, and for $\phi = 0.1$, and $\epsilon = 0.5$, (see *Figure 6b*). These values of our simulation parameters also correspond to the experimental conditions. Small volume fractions of chromatin ($\phi = 0.1$) correspond experimentally to large (hydrated) *Drosophila* nuclei (*Amiad-Pavlov et al., 2020*). We note that for these simulations a somewhat smaller self-attraction, $\epsilon = 0.5$, shows the angular separation of LAD and non-LAD chromatin as seen in the experiments, whereas for $\epsilon = 1$, we find a homogeneous peripheral distribution with mostly LAD domains near the lamina, similar to *Figure 6a*. Although the qualitative trends in the simulations and experiments are indeed similar, it would be interesting for new experiments to more quantitatively test our predicted transitions in the state diagram of *Figure 5*. For example, in vitro experiments could control the amount of dehydration (changing the chromosome fraction $\phi$ with fixed chromosome mass) by spreading cells on surfaces (*Guo et al., 2017*; *Adar and Safran, 2020*).

To focus on the essential physics of chromatin self-attraction vs. chromatin-lamina interactions, we have simplified the chromatin self-attractions to be the same for the euchromatin (EC) and heterochromatin (HC) domains. This is appropriate when the chromatin-aqueous phase interaction and the lamin-chromatin interaction are larger than the differences in the interactions of euchromatin and heterochromatin. Our model is the minimal one required to understand how the chromatin separates from the aqueous phase in peripheral and central organization. Although, previous models explicitly accounted for different interactions of euchromatin and heterochromatin domains, they did not consider the various types of nuclear-scale chromatin/aqueous phase/laminar organization and typically focused only on conventional chromatin organization. In addition, in the Appendix 7, we have discussed simulation results that show coexistence (micro phase separation) of more condensed and less condensed regions of chromatin due to different self-attractions of the LAD and non-LAD regions. This is analogous to the observed differences in concentration of chromatin in heterochromatin and euchromatin. Those results also show that peripheral organization cannot be achieved if difference in interaction of the LAD and non-LAD regions is too large. This justifies the original model presented in the main text in which differences in the self-attraction of the LAD and non-LAD regions are neglected, as a first approximation. But at the next order of approximation where we include differences in the self-attraction of different chromatin regions, the simulations show micro phase separation.

## Materials and methods

### Modeling the stretching energy and bending energy of the chromatin chain

The stretching energy of the chromatin chain is calculated as:

$$E_s^{\text{chrom}} = \sum_i k_s (\mathbf{r}_i - 2a)^2 \tag{4}$$

where $\mathbf{r}_i$ is the distance between $i^{th}$ and $(i+1)^{th}$ beads. $k_s$ and $2a$ are respectively, the spring constant and equilibrium distance between two neighboring beads.

The chromosome has a persistence length of 2 beads (1.2 kb) which is based on a previous estimate (*Marko and Siggia, 1997*) of a persistence length of 1–2 kb for interphase chromatin. Since the bending energy determines the persistence length, this estimate allows us to write the bending energy of chromosome as:

$$E_b^{\text{chrom}} = \sum_i k_b (1 - \cos \theta_i) \tag{5}$$

where $\theta_i$ is the relative angle between two neighboring beads, $k_b$ is bending stiffness which is taken as $l_p k_B T / \sigma$. Here $l_p$ is persistence length, σ is diameter of the bead, $k_B$ is Boltzmann constant, and T is absolute temperature.

## Langevin dynamics

The system is simulated calculating the Brownian dynamics of the chromatin beads in which the aqueous solvent is treated implicitly by the average frictional force it exerts on the moving beads and by the deviations from this average, represented by a stochastic force. For the use in computer simulations, the discrete time (*Doyle and Underhill, 2005*; *Plimpton, 1995*) equation of motion of the beads in this model is given by the Langevin equation:

$$\mathbf{r}_i(t + \Delta t) = \mathbf{r}_i(t) - \frac{\Delta t}{\gamma m} \nabla_{\mathbf{r}_i} E_{tot}(t) + \sqrt{\frac{6 k_B T \Delta t}{\gamma m}} \xi_i(t) \qquad (6)$$

where $m$ is bead mass, $\Delta t$ is the time-step, γ is damping contact (related to the friction of the beads with the aqueous solvent) and ξ is the stochastic force (thermal noise applied by the implicit aqueous solvent to each bead) and has the standard statistical properties used in Brownian dynamics simulations of equilibrium systems (*Doyle and Underhill, 2005*). The LAMMPS package is used for our calculations (*Plimpton, 1995*).

## Initial conditions and equilibration

Our initial condition has the chromatin chain comprising both LAD and non-LAD domains (that are many beads long) that are mixed along the chain as indicated by the sequence data (*Ho et al., 2014*). The center of mass of the chain is initially at the center of the nucleus and the dynamics are determined by the numerical solution of the stochastic Langevin equation that in general, includes the forces that arise from the interactions of the chromatin beads with themselves and the LAD domains with the lamina, plus the stochastic forces due to the implicit solvent. Initially, we include only the excluded volume chromatin-chromatin interactions, LAD-lamin attractions with LJ interactions and allow a fraction ψ of LAD domains within a distance $r_b$ of the lamina to create/break bonds with lamin beads, via the spring model, allowing for both bond creation and breaking dynamics. After equilibration, we changed the chromatin-chromatin interactions to allow for self-attraction and let the system re-equilibrate, again using the LAMMPS package (*Plimpton, 1995*). The parameter values used in the simulations are summarized in *Table 1*.

## Experiment's details

Experimental documentation techniques, sample sizes etc. are described in our experimental paper (*Amiad-Pavlov et al., 2020*).

**Table 1.** Parameter values used in the simulations.

| Parameter | Description | Reduced unit | SI unit |
|---|---|---|---|
| $k_B T$ | Thermal energy | 1.0 | $4.1 \times 10^{-21}$J |
| $m$ | Bead mass | 1.0 | $10^{-21}$ kg |
| σ | LJ size parameter | 1.0 | 10 nm |
| $\epsilon$ | LJ energy parameter | $1.0 k_B T$ | $4.1 \times 10^{-21}$J |
| $r_c$ | Contact distance | $2.5\sigma$ | 25 nm |
| $k_s$ | Spring constant | $100 k_B T / \sigma^2$ | 0.41 Jm$^{-2}$ |
| $l_p$ | Persistence length | $2.0\sigma$ | 20 nm |
| $k_b$ | Bending stiffness | $2.0 k_B T$ | $8.2 \times 10^{-21}$J |
| τ | Damping time | $1.0 \times (3\pi\eta\sigma^3 / K_B T)$ | 2 µs |
| $\Delta t$ | Time step | $0.01\tau$ | 20 ns |

## Acknowledgements
SAS and TV are grateful for the support of the Kretner-Katz Foundation and the Weizmann-Curie grant, and SAS for the support of Perlman family foundation, the US-Israel Binational Science Foundation, and the Volkswagen Foundation for their support. We thank D Deviri, Omar Adame-Arana, Phil Pincus, and Michael Rubinstein for useful discussions.

## Additional information

### Funding

| Funder | Grant reference number | Author |
|---|---|---|
| Perlman Family Foundation | | Samuel Safran |
| Volkswagen Foundation | | Samuel Safran |
| United States - Israel Binational Science Foundation | | Samuel Safran |
| Weizmann Institute of Science | Weizmann Curie Research Grant 2021 | Talila Volk Samuel Safran |
| Weizmann Institute of Science | Weizmann-Krenter-Katz Grant Program 2019 | Talila Volk Samuel Safran |

The funders had no role in study design, data collection and interpretation, or the decision to submit the work for publication.

### Author contributions
Gaurav Bajpai, Conceptualization, Formal analysis, Validation, Investigation, Visualization, Methodology, Writing - original draft, Writing - review and editing; Daria Amiad Pavlov, Dana Lorber, Talila Volk, Conceptualization, Validation, Investigation, Writing - review and editing, Performed and analyzed the experiments; Samuel Safran, Conceptualization, Formal analysis, Supervision, Funding acquisition, Validation, Investigation, Visualization, Methodology, Writing - original draft, Writing - review and editing

### Author ORCIDs
Gaurav Bajpai (ID) https://orcid.org/0000-0003-3875-4599
Dana Lorber (ID) https://orcid.org/0000-0002-0635-8703
Talila Volk (ID) http://orcid.org/0000-0002-3800-2621
Samuel Safran (ID) https://orcid.org/0000-0002-0798-1492

### Decision letter and Author response
Decision letter https://doi.org/10.7554/eLife.63976.sa1
Author response https://doi.org/10.7554/eLife.63976.sa2

## Additional files
### Supplementary files
• Transparent reporting form

### Data availability
All data generated or analysed during this study are included in the manuscript and supporting files. Source data files are available on zenodo using the following link: https://zenodo.org/record/4772105.

The following dataset was generated:

| Author(s) | Year | Dataset title | Dataset URL | Database and Identifier |
|---|---|---|---|---|
| Bajpai G, Amiad Pavlov D, Lorber D, Volk T, Safran S | 2021 | Mesoscale phase separation of chromatin in the nucleus | https://zenodo.org/record/4772105 | Zenodo, 4772105 |

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

# Appendix 1

## Comparison of the simulations with the experimental images

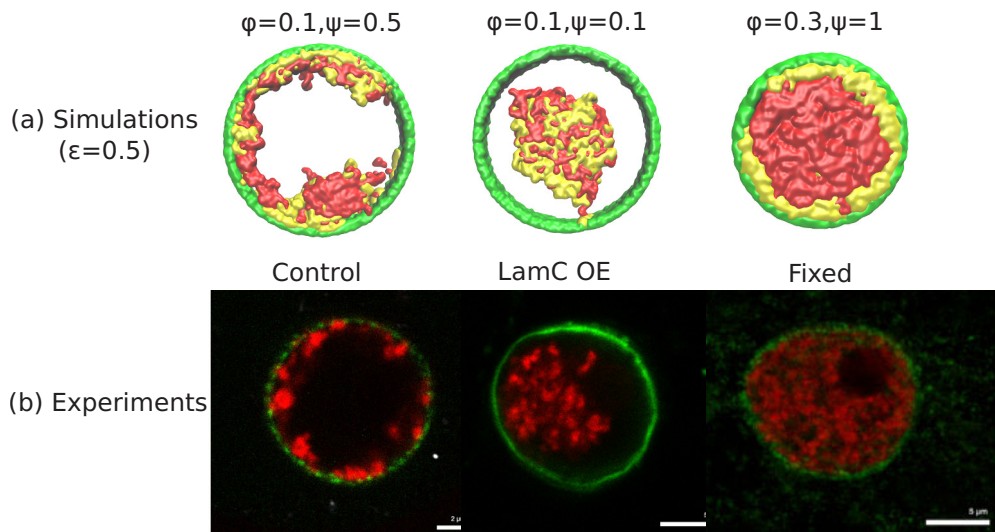

**Appendix 1—figure 1.** Comparison of the simulations with the experimental images. (**a**) Snapshots of simulations for chromatin-chromatin attractions $\epsilon = 0.5$. For relatively small volume fraction of chromatin ($\phi = 0.1$) and relatively high chromatin-lamina binding interactions ($\psi = 0.5$), the simulation shows peripheral organization of chromatin. For $\phi = 0.1$ (small) and $\psi = 0.1$ (small), we obtain central organization of chromatin (see Appendix 5 on the wetting droplet). For $\phi = 0.3$ (high) and $\psi = 1$ (high), the simulations show conventional organization of chromatin, where the entire nucleus is filled. (**b**) Different organization of chromatin obtained in the experiments: peripheral, central and conventional are seen in experiments in the control, lamin A/C overexpression and fixed nuclei respectively (*Amiad-Pavlov et al., 2020*).

# Appendix 2

## Unconfined chromatin chains: self-avoiding and self-attractive

Here, we demonstrate that in the absence of confinement, the chromatin chain of our simulation model obeys the known scaling behavior of long polymer chains in both the self-avoiding (good solvent) and self-attractive (poor solvent) scenarios. Self-avoiding chains are modeled by cutting off the LJ potential when the force goes to zero. As defined in the main text, this occurs for the cutoff $r_c = 2^{1/6} = 1.1225$. Self-attracting chains are modeled by including the attractive region of the LJ potential and the cutoff is set at $r_c = 2.5$. For a persistence length of two beads and a and LJ potential strength of $\epsilon = 1 k_B T$, we simulated self-avoiding chromatin chains of $N = 100, 200, 500$, and 1000 beads and self-attracting chains for additional values of $N$ up to 100,000 as shown in the figure (see *Appendix 2—figure 1*). To obtain the power law scaling from our simulations, we calculated the radius of gyration $R_g$ for these various values of $N$ and write.

$$R_g = \frac{l_p}{2} N_p^\nu \tag{7}$$

where $l_p$ is persistence length, $N_p$ is number of persistence length ($N_p = N/l_p$), and $\nu$ is the scaling exponent. For the self-avoiding case, a good fit is found with $\nu = 0.59$ and for the self-attracting case, $\nu \approx 1/3$ as expected (*Rubinstein and Colby, 2003*) from the theory of unconfined chains.

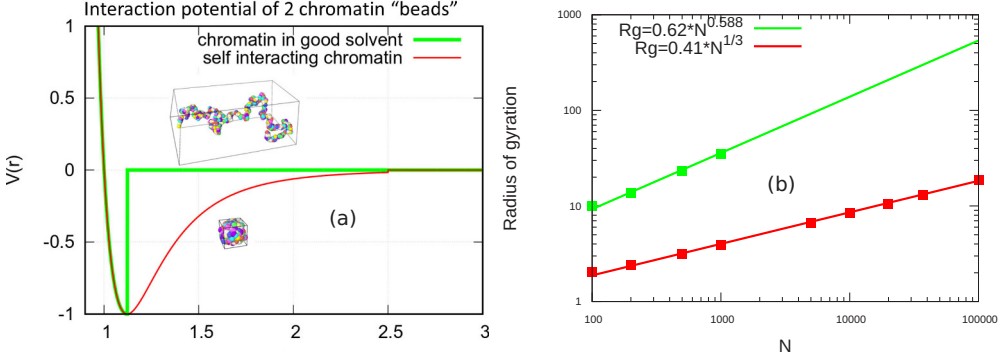

**Appendix 2—figure 1.** Scaling behavior of unconfined chromatin chains in good and poor solvent. (**a**) LJ potential as a function of the distance between two beads, with snapshot of simulations of chromatin chain having 500 beads. Snapshots show unconfined, self-avoiding chromatin in good solvent conditions for a cutoff distance $r_c = 2^{1/6} = 1.1225$ (green) and self-attractive chromatin in poor solvent conditions, for a cutoff distance $r_c = 2.5$. (**b**) Radius of gyration $Rg$ vs. the number of beads $N$ is calculated from the simulations of unconfined chromatin chain having $N = 100, ...100,000$ beads (green and red dots). The green and red lines are guides to the eye. They can be fit with the power laws shown, demonstrating the polymeric scaling in the unconfined case for both good and poor solvent conditions.

## Appendix 3

### Analysis of LADs sequence data of *Drosophila*

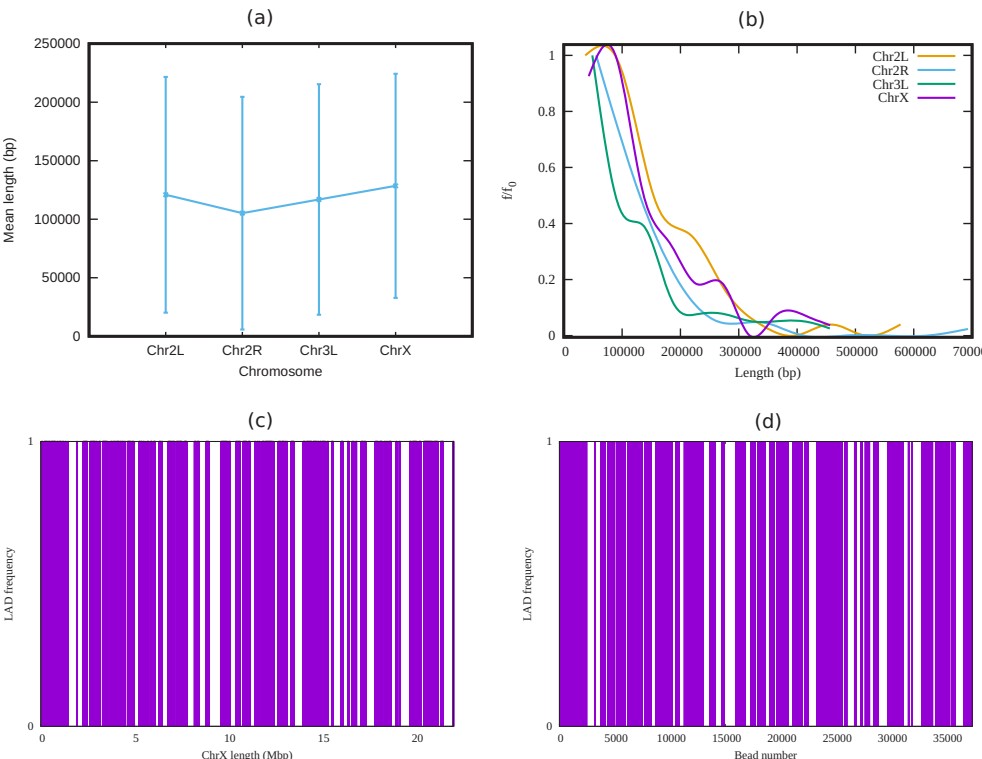

**Appendix 3—figure 1.** Analysis of LADs sequence data of *Drosophila*. (**a**) Mean cluster length of LAD for different chromosomes of *Drosophila* (*Ho et al., 2014*). (**b**) Length distribution of LAD for different *Drosophila* chromosomes. In both subfigures (**a**) and (**b**), the vertical bars represent the standard deviation (SD). (**c**) An alternating distribution of LAD (violet) and non-LAD (white) along the 22.4 Mbp regions of chromosome X (ChrX). (**d**) Same LAD distribution used in our coarse-grain model with a chromatin chain of 37,333 beads. From the sequence data it is clear, LAD regions in *Drosophila* consist of $\approx 150$ beads, and not single ones.

## Appendix 4

### Alternate simulations for random LAD sequences

To demonstrate that our simulation results do not depend on any specific LAD sequence data, we also generated LAD distributions using a Monte-Carlo method. We considered two cases for LAD binding to lamin: (i) Randomly distributed as single beads, (ii) Exponentially distributed cluster with a mean of 150 beads. (i) LAD randomly distributed as single beads: In this method, we considered a fraction $f$ of LAD beads within the chromatin chain that can bond to the lamin and distributed these randomly along the chain as single beads. (ii) LAD are exponentially distributed in clusters: In this method, we used basic knowledge of the experimental sequence data for *Drosophila* that indicates that LAD are distributed along the chromatin chain in an exponential manner, in regions with a mean length of $\approx 150$ beads. From this distribution, we used Monte-Carlo method to generate LAD regions along the chromatin chain. We first generated LAD clusters for a fraction $f$ of chromatin beads from an exponential distribution with a mean length 150 beads. We then arranged these clusters in descending order ($m1 > m2 > m3....$ where $m1, m2, m3$ are the number of beads in each cluster). Using a Monte-Carlo method, we first randomly choose a position ($1...N$ where $N$ is the total number of beads in the chain) for the cluster of size $m1$. Next, we choose a position for the cluster of $m2$ LAD beads, making sure that it does not overlap with the first cluster of $m1$ beads. After placing the $m1$ and $m2$ clusters in the chromatin chain, the $m3$ cluster can be placed between these two only if there is enough space between the two cluster to fit $m3$ beads. We repeat the process until the sum of all the clusters is equal to $Nf$ ($m1 + m2 + .... = Nf$).

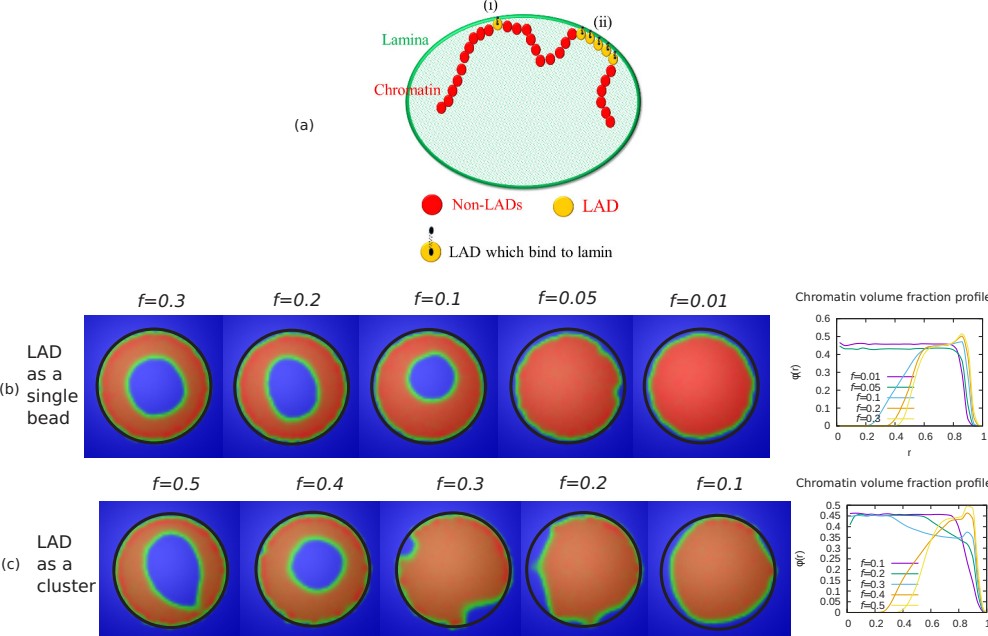

**Appendix 4—figure 1.** Alternate simulations for random LAD sequences. (**a**) Schematic diagram describing our coarse-grained model of chromatin-lamina interactions. Two cases for LAD binding to lamin: (i) Randomly distributed as single beads, (ii) Exponentially distributed cluster (many beads). Chromatin concentrations and local chromatin volume fraction profiles for $\phi = 0.3$ and $\epsilon = 1$ are shown when LAD binds to lamin as a (**b**) single bead (**c**) cluster.

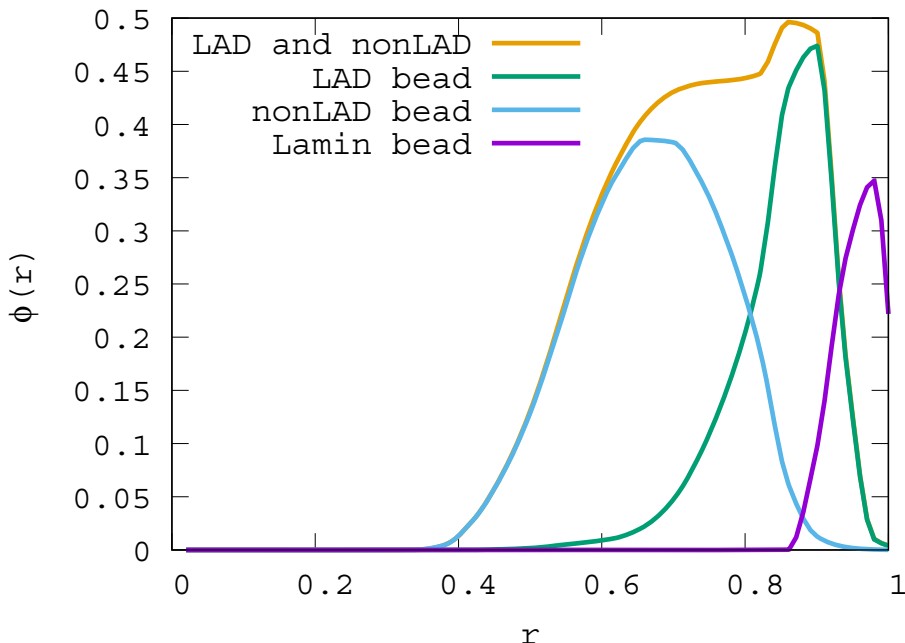

**Local volume fraction profiles of LAD, non-LAD, and lamin beads**

**Appendix 4—figure 2.** Local volume fraction profiles of LAD, non-LAD, and lamin beads. For $\phi = 0.3, \epsilon = 1, f = 0.5$, the local volume fraction profiles of LAD, non-LAD and lamin beads are shown. The graph shows the location of each of these bead types in the spherical volume where $r = 0$ is the sphere center and $r = 1$ is the sphere surface.

# Appendix 5

## Central organization and wetting droplet

From simulations of the chromatin for small values of $\psi$ (fraction of LAD which can bond to the lamin), we found a central organization of chromatin with several 'arms' (see 'arms' in *Appendix 5—figure 1a*) that contact the lamina. This occurred when the unbinding probability (related to the bond breaking as explained below) was relatively small and is thus a kinetic effect. Nevertheless, the experiments indicate such configuration. Since each LAD domain is rather large ($\approx 150$ beads and not a single 'monomer' of the chain), once it binds to the lamina, unbinding – which must involve the near-simultaneous detachment from the lamina of all 150 beads – occurs with low probability. These conditions resulted in the 'arms' that resembled the experiment (see *Appendix 5—figure 1b*). Different LAD domains will randomly contact the lamina at different positions, resulting in several 'arms'. If we greatly increase probability of unbinding, then even if an entire arm randomly binds, it can unbind and the system can reach the equilibrium of an approximately wetting droplet that contacts the lamina in one region and not via many 'arms'. Thus, this is a matter of time scales and the irreversibility of the LAD-lamin binding that determines whether the chromatin in the weak-binding limit, shows central localization with several bound 'arms' or whether it equilibrates to the wetting droplet. We recall that these effects refer to LAD-lamin binding as opposed to the physical chromatin-chromatin attractions, since LAD-lamin attractions are mediated by specific binding proteins. In *Appendix 5—figure 1a*, we show 3D snapshots of simulations for $\psi = 0.1, \phi = 0.1, \epsilon = 1$. All the configurations show central organization of chromatin. The difference between the first two from the left is in the detachment kinetics of the LAD domains; we control this by varying the distance $r_b$ between the LAD and the lamin at which the bond between them (biophysically mediated via BAF and other proteins not explicitly included in our simulation) breaks; that is, the spring energy representing the bond, abruptly goes to zero when the bond is stretched to a distance larger than $r_b$. In the 'wetting droplet' configuration (second from left) we take $r_b = 1.1225$, so that bond breaking happens much more easily than in the simulations (first from left) that show several 'arms' where $r_b = 2.5$ and the LAD-lamin distance can fluctuate over a larger range and still remain bonded. Thus, for $r_b = 1.1225$, the chromatin equilibrates into a 'wetting droplet' with the relatively simple geometry shown here with no 'arms' that are 'stuck' to the lamina. The third picture from the left shows the chromatin localization in the center with neither 'arms' nor 'wetting droplet' configurations for the case where no bonds are formed between the LAD and the lamin. The center of mass of the chromatin can be found anywhere within the nuclear volume.

(a) Simulations

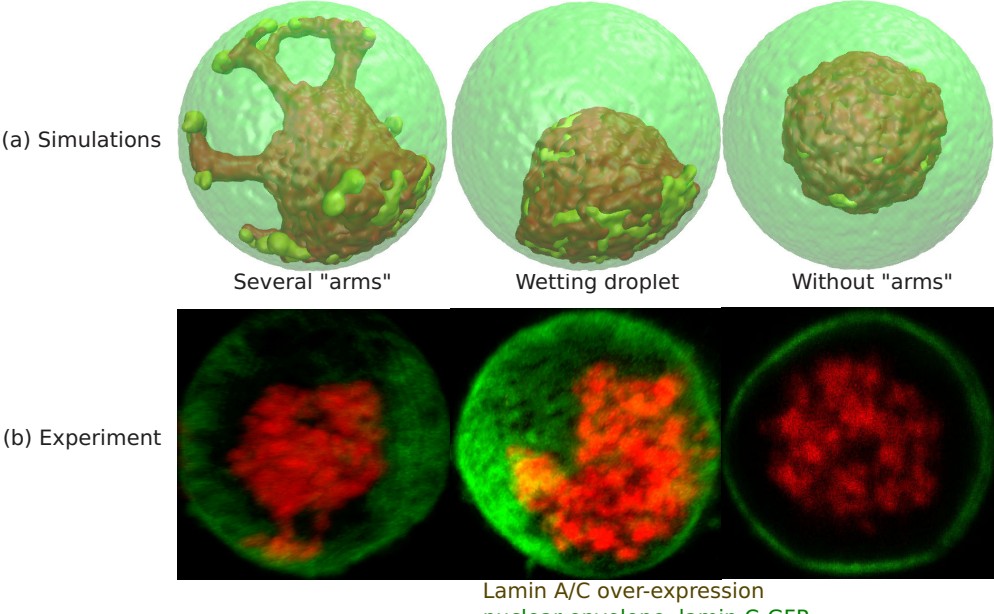

Several "arms"     Wetting droplet     Without "arms"

(b) Experiment

Lamin A/C over-expression
nuclear envolope, lamin C-GFP

*Appendix 5—figure 1 continued on next page*

*Appendix 5—figure 1 continued*

**Appendix 5—figure 1.** Wetting droplet and central chromatin organization for relatively weak LAD-lamina interactions. (**a**) 3D snapshots of simulations showing central organization of chromatin with/without 'arms' as well as the wetting droplet. (**b**) Experiments also suggest central organization of chromatin with/without 'arms' as well as the wetting droplet for the case of Lamin A/C overexpression (*Amiad-Pavlov et al., 2020*).

# Appendix 6

## Comparison of chromatin contacts in peripheral, central, and conventional organization

In Hi-C experiments, cells are fixed by formaldehyde which has been reported (*Li et al., 2017*; *Maeshima et al., 2020*) to lead to cell shrinkage; thus, the cell and nuclear volume of fixed cells can be considerably smaller than those of live, intact cells (by a factor of up to three in our experiments) (*Amiad-Pavlov et al., 2020*). It is thus of biological interest to compare the chromatin contact maps for cells with different chromatin volume fractions. Contact maps, as typically measured in Hi-C experiments, quantify the contacts between pairs of chromatin regions in cells. In the simulations presented here, we compare the contact properties of chromatin in nuclei with different volume fractions of chromatin, that correspond to fixed (conventional organization) and live (peripheral/central organization) cells. We calculated the contact map from the simulations whose snapshots are shown in *Appendix 1—figure 1a*. We define contact between two beads of chromatin chain when their distance is $\leq 1.5\sigma$. In *Appendix 6—figure 1a*, we compare the contact maps of central ($\psi = 0.1, \epsilon = 0.5, \phi = 0.1$) and peripheral ($\psi = 0.5, \epsilon = 0.5, \phi = 0.1$) organization of chromatin at 50 kbp resolution. We conclude that the map of central chromatin has more distal contacts (color) compare to peripheral chromatin. We know from previous studies that condensed chromatin shows less distal contact compared with open chromatin in nucleus (*Naumova et al., 2013*). This means that in our model, the peripheral chromatin corresponds to a more condensed state compared with central organization. Next, in *Appendix 6—figure 1b*, we compare the contact maps of conventional ($\psi = 1, \epsilon = 0.5, \phi = 0.3$) and peripheral ($\psi = 0.5, \epsilon = 0.5, \phi = 0.1$) chromatin. Our result show that both peripheral and conventional are in a relatively condensed state compared with central organization. To understand how chromosomes mix with time, we mark our long chromatin chain of 22Mpb with 2Mbp regions. We define contacts within these 2 Mbp chromatin regions as local contacts and contacts between the different 2 Mbp chromatin regions as distal contacts (*Chiang et al., 2019*). If we roughly think of each marked section as a separate chromosome, local contact may correspond to contact within a chromosome and distal contact may correspond to contact between chromosomes. We then calculated the open chromatin index (OCI) which is simply equal to ratio of the averaged sum (over all segments) of distal contact and averaged sum of local contact. In *Appendix 6—figure 1c*, we show the OCI for central, peripheral and conventional chromatin. The OCI increases with time for central chromatin and relatively quickly saturates to a constant value for conventional and peripheral. Our result shows that these different regions (roughly corresponding to different chromosomes) do not mix when their interaction with the lamin is high ($\psi \geq 0.5$) and remain in their original regions (analogous to territories) within the nucleus. To understand the nature of this packaging, we computed the contact probability ($P(s)$) along the length ($s$) of our model of chromosome X. This shows a power law decay with $P(s) \propto s^{-1}$, corresponding to a fractal globule where the various segments have not mixed (*Mirny, 2011*). In *Appendix 6—figure 1d*, each of the various chromatin organizations shows fractal globule behavior. However, since the OCI of central chromatin (pre-wetting droplet) increases with time, its fractal globule nature may evolve into equilibrium globule after a much longer time. Such dynamics are outside the scope of this paper.

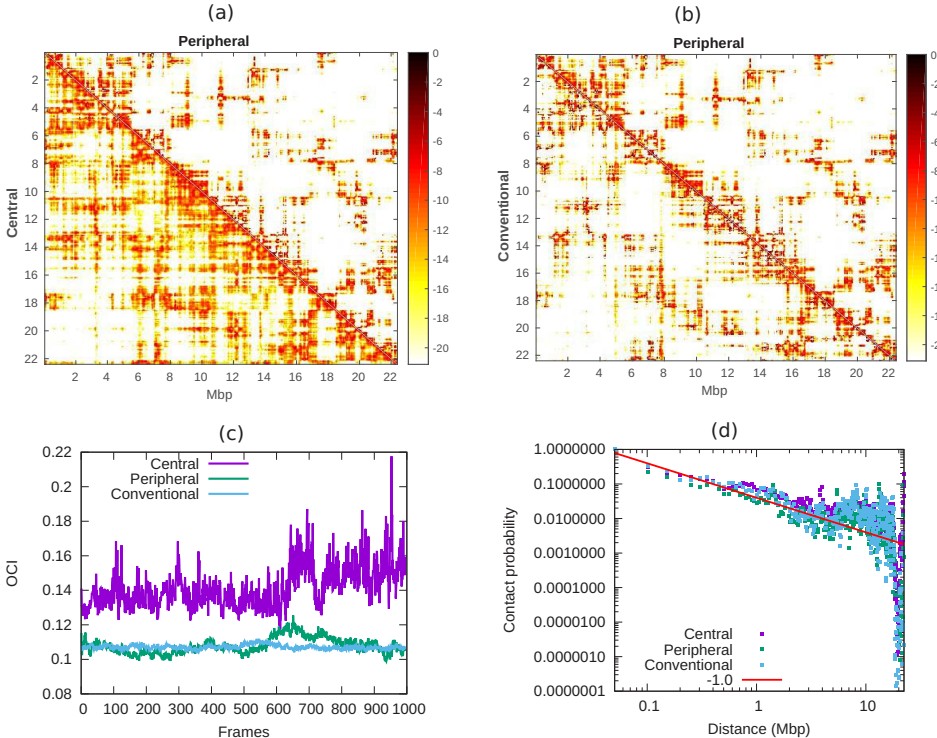

**Appendix 6—figure 1.** Comparison of chromatin contacts in peripheral, central, and conventional organization. (**a,b**) Contact maps from simulations, of different 50-kbp bins, for different chromatin organization. The color scheme varies from black to white, representing high to low contact counts ($\log(P_{ij})$). (**a**) Comparison of contact map of central (lower diagonal) and peripheral (upper diagonal) chromatin organization. (**b**) Comparison of contact map of conventional (lower diagonal) and peripheral (upper diagonal) chromatin organization. (**c**) Open chromatin index (OCI) plotted with respect to time (simulation frames) for central, peripheral, and conventional organization chromatin. (**d**) Contact probability as a function of contour distance for central, peripheral, and conventional organization chromatin, calculated from the simulations (dots). The red line is a guide to the eye indicating a power-law behavior, which suggests the fractal globule packing nature.

# Appendix 7

## Phase separation of euchromatin-like and heterochromatin-like regions in conventional chromatin packing within the nucleus

In our previous results, we considered the case where the self-attractions of the LAD and non-LAD beads were identical; the only difference between them was that only the LAD were attracted (bonded) to the lamin. We studied the changes in the chromatin organization as we reduced the fraction of bonds between LADs and lamina ($\psi$), which, for small enough $\psi$, results in a transition from peripheral to central (wetting droplet) organization. Here, we consider the effects of a difference in the self-attraction of the LAD and non-LAD beads with each other as a model of the different self-attractions within heterochromatin and euchromatin. However, we realize that there are other aspects of hetero and euchromatin, such as their different post-translational modifications that are not explicitly included here. We modelled the non-LAD beads as a self-avoiding (polymer in good solvent) and the LAD beads as a self-attracting (polymer in poor solvent). Due to the self-avoiding non-LAD regions, the chromatin fills the entire nucleus and shows conventional organization (see *Appendix 7—figure 1*). We now change the bonding interactions between the LAD and lamina beads via a reduction in the parameter $\psi$. *Appendix 7—figure 1* shows phase separation of the non-LAD and LAD regions for different values of $\psi$. The snapshots of the simulations (see *Appendix 7—figure 1a*) indicate that for $\psi = 1$, the LAD beads (yellow) are distributed in the periphery and separated from the non-LAD (red). For $\psi = 0.5$, we can see mixing of the LAD and non-LAD beads. When $\psi = 0$, all the LAD beads are distributed in the center but due to the different self-attractions used in the simulations discussed in this section, are phase separated from the non-LAD beads. In *Appendix 7—figure 1b*, we show the chromatin concentration profile where high chromatin concentration is represented by red, low concentration by green and aqueous phase by blue. These demonstrate the existence of A/B like compartments that may correspond to the non-uniform distribution of chromatin in different cells (*Falk et al., 2019*). The central panel of *Appendix 7—figure 1b* which shows a more uniform filling of the nucleus by the non-LAD beads may correspond to the patchy 'marshland' measured for chromatin in aqueous phase *Cremer et al., 2015*. The results in this section show that different chromatin packing concentrations can be obtained by a combination of different attractions and LAD lamin attraction. While not a complete model of hetero and eu chromatin, they do have certain features in common with the different packings commonly associated with these two chromatin types.

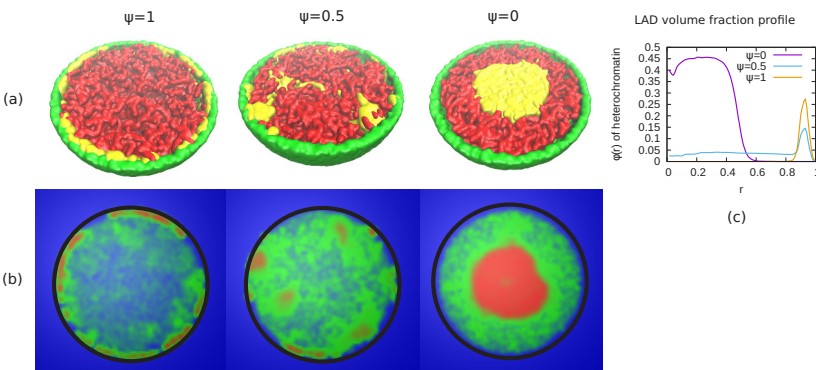

**Appendix 7—figure 1.** Phase separation of euchromatin-like and heterochromatin-like regions in conventional chromatin. (**a**) Snapshots of simulations for different $\psi$ are shown. In figure, LAD beads (yellow) are self attractive chain whereas non-LAD beads (red) are self-avoiding. This modeling results in a stronger separation of the two types of beads into spatially distinct regions. (**b**) Chromatin concentrations for different $\psi$ are shown. High (red) and low (green) concentrations of chromatin with aqueous phase (blue) shows A/B compartments like (non-uniform) distribution of chromatin. (**c**) Local volume fraction of the LAD beads shows peripheral organization of those regions for $\psi = 1$, a uniform distribution for $\psi = 0.5$ and central organization for $\psi = 0$.

