## [Decision Letter]

**Acceptance summary:**

The authors study theoretically chromatin distribution in the nucleus. This is a very well-written study that identifies the physical variables underlying observed nuclear phase separation. It is of broad interest for understanding chromatin organization.

**Decision letter after peer review:**

Thank you for submitting your article "Mesoscale phase separation of chromatin in the nucleus" for consideration by *eLife*. Your article has been reviewed by 3 peer reviewers, one of whom is a member of our Board of Reviewing Editors, and the evaluation has been overseen by K VijayRaghavan as the Senior Editor. The following individual involved in review of your submission has agreed to reveal their identity: Helmut Schiessel (Reviewer #2).

The reviewers have discussed the reviews with one another and the Reviewing Editor has drafted this decision to help you prepare a revised submission.

As the editors have judged that your manuscript is of interest, but as described below there are some doubts on whether your work goes sufficiently beyond your Ref.[12] of the manuscript to merit publication in *eLife*.

You study theoretically the effect of nuclear volume and the interaction of chromatin with the lamina on the distribution of chromatin within the nucleus using a simple polymer model. In your model, you also consider self-attraction of the chromatin. You find that depending on the relative strength of these effects, the chromatin is either homogeneously distributed within the nucleus or is localized peripherally. An intermediate state with the chromatin "partially wetting" the nuclear lamina is also observed. These states of chromatin distribution were also observed in *Drosophila* nuclei. This is a very well-written study that presents. The most important finding being that of the effect of the solvent quality on phase separation. However, the fact that the configuration of a spherically confined polymer can steered in the manner shown by directly manipulating the relative self-interaction versus the affinity of (parts of) the chain to the boundary is not really surprising. Also it is not clear, whether the difference between eu- and heterochromatin can indeed be neglected in this context.

If you decide to submit a revised version to *eLife*, please, make sure to comment in particular on why you think that the results presented here and that go beyond Ref.[12] merit publication in *eLife* rather than a more physics oriented journal like Soft Matter.

Essential revisions:

1. Why do you need a second potential in addition to the LJ potential after bond formation between an LAD bead and a lamina bead?

2. Are the effects of fixation well described by changing the chromatin volume fraction alone?

3. The paragraph on page 8 about bond formation is a little difficult to follow. Could you clarify the probability for bond breaking *P_break_*}. One would expect to find in the exponent the difference between the elastic energy at σ and at *r_b_*.

4. Earlier work in *C. elegans* (e.g. Meister P et al. Genes Dev. (2010)) seems to suggest that the specific affinity of heterochromatin to the lamina is important to understand the global structure of the nucleus. The authors definitely appear aware of this issue, by explicitly stating that their model holds whenever the generic interactions between chromosome and lamins are more important than the e.g. eu/hetero cross-interactions. However, in the discussion they appear to want to have it both ways in the end, by appealing (without further references) to a correlation between the occurrence of LAD domains and the eu/hetero nature of regions of the chromosome. Could you clarify this issue?

---

## [Author Response]

You study theoretically the effect of nuclear volume and the interaction of chromatin with the lamina on the distribution of chromatin within the nucleus using a simple polymer model. In your model, you also consider self-attraction of the chromatin. You find that depending on the relative strength of these effects, the chromatin is either homogeneously distributed within the nucleus or is localized peripherally. An intermediate state with the chromatin "partially wetting" the nuclear lamina is also observed. These states of chromatin distribution were also observed in *Drosophila* nuclei. This is a very well-written study that presents. The most important finding being that of the effect of the solvent quality on phase separation.

We thank the reviewers for their positive approach regarding our article; we are also grateful to the reviewers for their insightful comments and questions that have helped us to improve the manuscript. Below we address the reviewers’ concerns in detail. Based on the referee comments, we have done new simulations and have revised the manuscript to address all the concerns. The changes in the manuscript have been marked in red in the revised manuscript.

However, the fact that the configuration of a spherically confined polymer can steered in the manner shown by directly manipulating the relative self-interaction versus the affinity of (parts of) the chain to the boundary is not really surprising.

We believe that our paper is important for biologists because it presents a new paradigm that the chromatin can be (depending on hydration) phase separated in the nucleus. This is the first time that, to our knowledge, chromatin phase separation is discussed based on experimental data. That is why we submitted the paper to *eLife* and not to a physics journal, proper. The observations, model simulations and explanations change the biological paradigm and show that the competition of the surface and bulk interactions, can indeed be fine-tuned to lead to the transitions between peripheral, conventional and central (wetting droplet) organization. Our paper provides the conceptual and modeling basis for these transitions which are indeed observed in the experiments as we discuss in the paper.

Moreover, we present a guide for further experiments that can map the global state diagram showing the transitions between conventional, peripheral and central (wetting droplet) organization of chromatin. An additional interesting observation, simulation result and explanation involves the wetting droplet (in equilibrium, as shown in Appendix 5–figure 1) as well as the non-equilibrium droplet with arm-like extensions that are kinetically “stuck” to the lamin layer (also shown in Appendix 5–figure 1); these are not commonly used in biology or biophysics. In our simulations in Appendix 5, we show that these different situations are kinetically determined.

Because chromatin organization can have profound effects on gene expression (see just below for the text we inserted in the paper), it is important to explain these effects and quantify them for biologists and biophysics even though some of the qualitative properties may not surprise experts in polymer physics, per se. That is precisely the reason we are submitting to *eLife*. In light of the referee comment, we have further explained the novelty and importance of the paper on page 3 of the Introduction where we added the following text: “In this paper we systematically examine how the interplay between chromatinlamina interactions, intra-chromatin interactions, and hydration affect the chromatin organization. […] Our simulation results also provide suggestions to experimental groups on how to vary the minimal number of parameters that will experimentally result in different modes of chromatin organization in vivo.”

Also it is not clear, whether the difference between eu- and heterochromatin can indeed be neglected in this context.

We justify our use of the same self-attractions of eu and hetero chromatin, as a first approximation only, from the experiments themselves. The experimental figure of the peripheral organization shown in Figure 6(c), indicates that both heterochromatin and euchromatin are found in the peripheral part of the nucleus. They both separate from the aqueous phase in the center. This suggests that the chromatin attractions driving this phase separation are to a first approximation, similar for heterochromatin and euchromatin. Of course to the next approximation, the self-attraction of heterochromatin and of euchromatin differ and that is why the experiments imaged alternating regions of heterochromatin and euchromatin along the angular direction in the nuclear periphery (see Figure 6(c)). Our simulations show that this can indeed be the case when the LAD interactions with the lamin are larger than the smaller differences between the self-attraction of euchromatin vs. heterochromatin (see Figure 6(b)). We explained this in the paper on page 10 of the Discussion part.

Based on the referee comments, we have added new simulation results to the paper that do take into account different self-attraction for the LAD compared to the self-attraction of the non LAD regions. Note that we are aware that there are other differences between heterochromatin and euchromatin such as histone modifications. These new simulations are not meant to replicate hetero and euchromatin in detail, but rather to show that micro phase separation within the chromatin (as opposed to chromatin/aqueous phase separation) can occur even for different self-attractions of the LAD and non-LAD regions. This is analogous to eu and heterochromatin whose detailed study is outside the scope of our paper. We have done this in the limiting case that the attractions of the LAD regions (as an analogy of heterochromatin) are larger than the non LAD regions (as an analogy of euchromatin). In Appendix 7, motivated by previous models, we considered the non-LAD regions as polymers in good solvent and the LAD regions as polymers in poor solvent [3]. These new simulation results show phase separation of the LAD and nonLAD regions for the relatively high volume fraction regime where the overall chromatin is conventionally packed and not phase separated from the aqueous phase (see the new Appendix 7 and Appendix 7–figure 1 in revised manuscript). Our new results also demonstrate that peripheral organization cannot be achieved if the difference in interactions of the self-attractions of the LAD and non-LAD is large. This justifies our original model as presented in the main text in which differences in the self-attraction of the LAD and non-LAD are neglected to a first approximation. The primary effect is then phase separation of chromatin from the aqueous phase as observed. Our new simulations demonstrate that the largest interactions are the chromatin-chromatin self-attractions and that the differences in those attractions due to other modifications (such as LAD and non-LAD) are secondary.

We have referred to these new results of Appendix 7, in the main text on page 5, where we added the following text:

“In addition, in the Appendix 7, we also consider the case in which interactions between two LAD beads are more strongly attractive compared with the interactions of two nonLAD beads or a LAD non-LAD pair. […] But at the next order of approximation where we include differences in the self-attraction of different chromatin regions, the simulations show micro phase separation.”

If you decide to submit a revised version to eLife, please, make sure to comment in particular on why you think that the results presented here and that go beyond Ref.[12] merit publication in eLife rather than a more physics oriented journal like Soft Matter.

We believe that paper is important for biologists because it presents a new paradigm that the chromatin can (depending on hydration) phase separate from the aqueous phase in the nucleus. To our knowledge, this the first time that chromatin phase separation is analyzed using simulations and discussed physically in comparison with experimental data. This evidence changes the biological paradigm and shows that the competition between the surface (lamin) and bulk chromatin interaction can indeed be fine tuned in both simulations and experiments to result in transitions between peripheral, conventional and central organization. Those transitions are indeed observed in the experiments as we discuss. Moreover, we present a guide to further experiments that can map the global state diagram of conventional, peripheral and central (wetting droplet) organization of chromatin. Because the organization can have profound effects on gene expression, it is important to explain these effects and quantify them for biologists and biophysicists even though the qualitative polymer properties may not surprise experts in polymer physics. That is precisely the reason we are submitting to *eLife*.

Based on the referee remarks we have further highlighted the biological implications of our results in a manner that relates to experiment. We have now included the results of additional calculations that relate our predictions of phase separation to the Hi-C contact maps of chromatin, which are of great interest to biologists. In particular, to relate the new paradigms of chromatin organization to the contact maps, we calculated those contact maps from our simulation results and compared the maps of peripheral, conventional and central organization (see Appendix 6 and Appendix 6–figure 1 in revised manuscript). Our new results show that chromatin in peripheral organization is condensed (in the sense of Hi-C maps) in a manner that is comparable to conventional organization but is more condensed (in the Hi-C sense) than central organization (Appendix 6–figure 1(a) and (b)). We have also shown that the lamin attraction prevents peripherally organized chromosomes from mixing and help chromosomes remain in well-define regions (analogous to territories) (see Appendix 6–figure 1(c)). We have also shown that similar to conventional organization, peripheral chromatin is also organized in a fractal globule fashion (see Appendix 6–figure 1(d)).

We have summarized the new results on the Hi-C contact map (presented in detail in the new Appendix 6) and how they impact the various chromatin concentration profiles in the main paper of the paper on page 10, where we added the following text:

“In Appendix 6 and Appendix 6–figure 1 we calculated contact map from our simulation results (whose snapshots are shown in Appendix 1–figure 1(a)) and compared the contact maps of peripheral, conventional and central organization. […] We have also shown that like conventional, peripheral chromatin is also organized in a fractal manner (see Appendix 6–figure 1(d)).”

Essential revisions:1. Why do you need a second potential in addition to the LJ potential after bond formation between an LAD bead and a lamina bead?

We used harmonic potential in addition to the LJ potential for following reasons:

i. The short-range LJ attraction (ϵ = 1*k_B_T, r_c_*= 2.5*σ*) was not strong enough to have localize LAD beads in the periphery. Even for cases where the LJ simulation parameters are larger (but still reasonable) and should favor peripheral organization, this potential was not enough to localize in periphery. That is accomplished by a more specific bonding model which introduces a strong energy penalty for changing the bond length. The main difference with the LJ interaction (which causes “polymer adsorption” but not bonding) is that once a bond is formed, there can be a strong energy penalty for changing the bond length (see below for the discussion of bond breaking).

ii. The LAD regions are initially uniformly distributed within the nucleus. The LJ attraction allow the LAD regions to kinetically become localized at the periphery where the bond formation further stabilizes their connection to the lamin. Biologically, the anchor protein lamin B receptor (LBR) binds lamin B and acts as a tether between chromatin and the nuclear lamina. To mimic LBR protein binding, we used a bonding potential between lamin and some fraction of the LAD beads. We restricted only a fraction, *ψ*, of the LAD beads to bind to simulate the finite fraction of binding sites or binding proteins; those were changed by the experiments in which lamin A/C were overexpressed. The use of bonding and the restriction of the number of binding sites/proteins allows us to predict how the chromatin concentration profile changes as a function of the fraction *ψ* of bonds that can be formed (e.g., experimentally caused by the finite number of LBR proteins or available lamin binding sites). This corresponds to the experiments on overexpression of Lam A/C (see Appendix 1–figure 1). These qualitative changes in chromatin organization as a function of the fraction of bonds that can be formed, cannot be studied using the LJ interaction which results in polymer adsorption to the lamina but not bonding, which can be limited due to the bonding restrictions.

iii. In our model, the bond between lamin and LAD is dynamic and can break (see below for a discussion of the breaking probability). Also, a single LAD bead can form a maximum of one bond with a given lamin bead. So, to simulate the transition from peripheral to central, we need only change the breaking bond properties of the LAD and lamin (see Appendix 5 and Appendix 5–figure 1 in revised manuscript).

In light of the referee comment, we have further explained the importance of bonding potential in the paper on page 5 of the Model where we added the following text:

“These qualitative changes in chromatin organization as a function of the fraction of bonds that can be formed, cannot be studied using the LJ interaction which results in polymer adsorption to the lamina and not bonding, which can be restricted due to a finite number of binding sites or available binding proteins.”

2. Are the effects of fixation well described by changing the chromatin volume fraction alone?

It has been reported that tissue fixation can cause a reduction in both the cellular and nuclear volumes due to dehydration of the cell [4–6]. For example, in Hi-C experiments, cells are fixed by formaldehyde. It has been reported the fixation of cells with formaldehyde leads to cell shrinkage and the cell volume of fixed cells can be significantly smaller than the cell volume of living cells [7,8]. Also, our experiments measured the volume of live *Drosophila* larva nuclei as having an average value of 1183*µm*^3^, while fixed nuclei had an average volume of 381*µm*^3^ [9].

3. The paragraph on page 8 about bond formation is a little difficult to follow. Could you clarify the probability for bond breaking P_break_. One would expect to find in the exponent the difference between the elastic energy at σ and at r_b_.

We apologize for the confusion. The bond breaking kinetics are parameterized in a general manner by *P_break_*that does not require a detailed formulation of the interaction potential responsible for bonding, other than its curvature near the minimum, as we now explain here and in the revised text:

i. In Langevin simulations, per se, only forces are relevant.

ii. In the bond model the relevant force is due to a spring (bond) that is formed when*r* (which is LAD-lamin distance) is less than or equal to *r_b_*≡ 2.5*σ*. Changes in the LAD lamin distance results in a spring-like force that restores the bond to its optimal length of *r* = *σ* with a spring constant of 10*k_B_T/σ*^2.^

iii. The bond stabilization energy i.e. energy when *r* = *σ* enters only indirectly since the bond breaking probability (and not a detailed bond potential) is used by the LAMMPS software. (We have appropriately modified the original discussion in the paper; see below.)

iv. Two parameters enter to describe bond breaking in LAMMPS.

*a. P_break_*: Probability that over many time steps the bond will stochastically break.

*b. r_b_*: Bond breaking only occurs if *r* (the LAD-lamin distance) obeys *r > r_b_*. That means, when *r* is too small, bond breaking does not occur and the bond distance fluctuates due to the competition of thee thermal forces and the spring.

v. These two features allow us to describe bond formation and bond breaking withoutincluding a detailed attractive potential that describes how the bond is formed with all intermediate configurations.

vi. We have therefore taken *P_break_*= *e*^−*Ubond/kBT*^where we set *U_bond_*= 10*k_B_T*. This represents a fairly long-lived bond. We took *r_b_*= 2.5*σ*. For bonds whose formation/breaking is given by detailed balance, *U_bond_*is the energy gained upon formation of the bond (relative to the unbonded state). In Appendix 5 where we show the wetting droplet, we have compared a value of *r_b_*= 1.1225*σ* where the unbonding is fast so that the equilibrium wetting droplet (Appendix 5–figure 1(a)) is obtained. For *r_b_*= 2.5*σ*, the unbonding is slow and the system remains “stuck” in the non-equilibrium state with many arms as show in Appendix 5–figure 1(a).

In light of the referee comment, we have further explained the probability for bond breaking *P_break_*in the paper on page 5 of the Model section where we added the following text: “The difference between the LJ interactions of the lamin and LAD and the bonding energy is seen when one considers the detachment of the lamin and LAD. […] (In Appendix 5, we describe the effects of variations of the cutoff distance for bond breaking.) ”

4. Earlier work in *C. elegans* (e.g. Meister P et al. Genes Dev. (2010)) seems to suggest that the specific affinity of heterochromatin to the lamina is important to understand the global structure of the nucleus. The authors definitely appear aware of this issue, by explicitly stating that their model holds whenever the generic interactions between chromosome and lamins are more important than the e.g. eu/hetero cross-interactions. However, in the discussion they appear to want to have it both ways in the end, by appealing (without further references) to a correlation between the occurrence of LAD domains and the eu/hetero nature of regions of the chromosome. Could you clarify this issue?

First, we thank to referee for the reference [11]; we have now added this to the revised manuscript. In the revised manuscript (and in our comments to the referees and the paper revisions noted above), we have now clarified that the simulations relate to LAD and non-LAD regions of chromatin. These is no modeling of hetero and euchromatin as relates to histone modifications. However, in the new Appendix 5 we now present a study of a system where the attractions of the LAD and non-LAD beads are different.

This results in micro phase separation of those regions and is analogous to micro phase separation associated with eu and heterochromatin.

In the peripheral organization, the experiment image in Figure 6(c) shows that both heterochromatin and euchromatin are found in the peripheral part of the nucleus. They both separate from the aqueous phase in the center. This suggests that the chromatin attractions driving this phase separation are to a first approximation, similar for heterochromatin and euchromatin as we assumed in the main text. Of course in the next order of approximation, the self-attraction of heterochromatin and of euchromatin differ and that is why the experiments imaged alternating regions of heterochromatin and euchromatin along the angular direction in the nuclear periphery (see Figure 6(c)). In Discussion part, based on our simulation results (see Figure 6(b)), we suggest that this can be the case when LJ attraction between LAD-lamin and nonLAD-lamin are same but LAD-lamin also have bonding attraction.

**References:**

1. Jain N, Iyer KV, Kumar A, Shivashankar G (2013) Cell geometric constraints induce modular gene-expression patterns via redistribution of hdac3 regulated by actomyosin contractility. Proceedings of the National Academy of Sciences 110:11349–11354.

2. Thomas CH, Collier JH, Sfeir CS, Healy KE (2002) Engineering gene expression and protein synthesis by modulation of nuclear shape. Proceedings of the National Academy of Sciences 99:1972–1977.

3. Chiang M, et al. (2019) Polymer Modeling Predicts Chromosome Reorganization in Senescence. Cell Reports 28:3212–3223.e6.

4. Xie K, Yang Y, Jiang H (2018) Controlling cellular volume via mechanical and physical properties of substrate. Biophysical journal 114:675–687.

5. Windner SE, Manhart A, Brown A, Mogilner A, Baylies MK (2019) Nuclear scaling is coordinated among individual nuclei in multinucleated muscle fibers. Developmental cell 49:48–62.

6. Pawley J (2006) Handbook of biological confocal microscopy (Springer Science and Business Media) Vol. 236.

7. Li Y, et al. (2017) The effects of chemical fixation on the cellular nanostructure. Experimental cell research 358:253–259.

8. Maeshima K, Tamura S, Hansen JC, Itoh Y (2020) Fluid-like chromatin: Toward understanding the real chromatin organization present in the cell. Current opinion in cell biology 64:77–89.

9. Amiad-Pavlov D, Lorber D, Bajpai G, Safran S, Volk T (2020) Live imaging of chromatin distribution in muscle nuclei reveals novel principles of nuclear architecture and chromatin compartmentalization. bioRxiv.

10. Blake T, De Coninck J (2011) Dynamics of wetting and Kramers’ theory. The European Physical Journal Special Topics 197:249–264.

11. Meister P, Towbin BD, Pike BL, Ponti A, Gasser SM (2010) The spatial dynamics of tissue-specific promoters during *C. elegans* development. *Genes and development* 24:766–782.